# Synergistic celecoxib and dimethyl-celecoxib combinations block cervix cancer growth through multiple mechanisms

**Diana Xochiquetzal Robledo-Cadena** [1,2], **Silvia Cecilia Pacheco-Velázquez** [3], **Jorge Luis Vargas-Navarro** [4], **Joaquín Alberto Padilla-Flores** [4], **Rebeca López-Marure** [5], **Israel Pérez-Torres** [6], **Tuuli Kaambre** [7], **Rafael Moreno-Sánchez** [4,7] *, **Sara Rodríguez-Enríquez** [8] *

**1** Departamento de Bioquímica, Instituto Nacional de Cardiología Ignacio Chávez, Ciudad de México, México, **2** Posgrado en Ciencias Biológicas, Universidad Nacional Autónoma de México, Unidad de Posgrado, Edificio D, 1° Piso, Circuito de Posgrados, Ciudad Universitaria, Coyoacán, C.P. 04510, CDMX, México, **3** Center for Preventive Cardiology, Knight Cardiovascular Institute, Oregon Health & Science University, Portland, OR, United States of America, **4** Laboratorio de Control Metabólico, Carrera de Biología de la Facultad de Estudios Superiores Iztacala, UNAM, Los Reyes Ixtacala, Hab Los Reyes Ixtacala Barrio de los Árboles/Barrio de los Héroes, Tlalnepantla, México, **5** Departamento de Fisiología, Instituto Nacional de Cardiología Ignacio Chávez, Ciudad de México, México, **6** Departamento de Biomedicina Cardiovascular, Instituto Nacional de Cardiología Ignacio Chávez, Ciudad de México, México, **7** Laboratory of Chemical Biology, National Institute of Chemical Physics and Biophysics, Tallinn, Estonia, **8** Laboratorio de Control Metabólico, Carrera de Médico Cirujano de la Facultad de Estudios Superiores Iztacala, UNAM, Los Reyes Ixtacala, Hab Los Reyes Ixtacala Barrio de los Árboles/Barrio de los Héroes, Tlalnepantla, México

* sara.rodriguez@iztacala.unam.mx, saren960104@hotmail.com (SRE); rafael.moreno@iztacala.unam.mx (RMS)

## Abstract

### Objective

The synergistic inhibitory effect of celecoxib (CXB) and dimethyl-celecoxib (DMC) plus paclitaxel (PA) or cisplatin (CP) on human cervix HeLa and SiHa cells was assessed at multiple cellular levels in order to elucidate the biochemical mechanisms triggered by the synergistic drug combinations.

### Methods

The effect of CXB (5 µM)/CP (2 µM) or CXB (5 µM)/PA (15 µM) and DMC (15 µM)/CP (5 µM) or DMC (15 µM)/PA (20 µM) for 24 h was assayed on cancer cell proliferation, energy metabolism, mitophagy, ROS production, glycoprotein-P activity, DNA stability and apoptosis/necrosis.

### Results

Drug combinations synergistically decreased HeLa and SiHa cell proliferation (>75%) and arrested cellular cycle by decreasing S and G2/M phases as well as the Ki67 content (HeLa) by 7.5–30 times. Cell viability was preserved (>90%) and no apparent effects on non-cancer cell growth were observed. Mitochondrial and glycolytic protein contents (44–95%) and $\Delta\Psi m$ (45–50%) in HeLa cells and oxidative phosphorylation and glycolysis fluxes (70–90%)

**Data Availability Statement:** All relevant data are within the manuscript and its Supporting Information files.

**Funding:** The present work was partially supported by grants from CONAHCyT-México to DXRC (No. 814560), CONAHCyT-México (No. 283144) and PAPIIT, DGAPA-UNAM, México to SRE (No. IA201823), and CONAHCyT-México (No. 6379) and National Institute of Chemical Physics and Biophysics (NICPB), Tallinn, Estonia Institutional Development Fund to RMS. There was no additional external funding received for this study. The funders had no role in study design, data collection and analysis, decision to publish, or preparation of the manuscript.

**Competing interests:** The authors have declared that no competing interests exist.

in HeLa and SiHa cells were severely decreased, which in turn promoted a drastic fall in the ATP supply (85–88%). High levels of mitophagy proteins in HeLa cells and active mitochondrial digestion in HeLa and SiHa cells was observed. Mitochondrial fission and microtubule proteins were also affected. Intracellular ROS content (2–2.3-fold) and ROS production was stimulated (2.3–4 times), whereas content and activity of glycoprotein-P (45–85%) were diminished. DNA fragmentation was not observed and apoptosis/necrosis was not detected suggesting that cell death could be mainly associated to mitophagy induction.

## Conclusions

CXB or DMC combination with canonical chemotherapy may be a promising chemotherapy strategy against cervical cancer growth, because it can selectively block multiple cell processes including inhibition of energy pathways and in consequence ATP-dependent processes such as cell proliferation, glycoprotein-P activity, ROS production and mitophagy, with no apparent effects on non-cancer cells.

## Introduction

Although too often associated with unwanted adverse side effects and the development of drug resistance, chemotherapy remains the gold standard of cancer treatment. Accordingly, combination chemotherapy is an emerging approach seeking to overcome the limitations associated with single-drug treatments.

A central aim in the development and evaluation of drug combinations is to achieve inhibitory synergy by demonstrating that the combined effect is significantly greater than that expected from the additive algebraic effects of individual drugs. An additional advantage of a synergistic drug combination may be that such approach may allow for lowering dosage, leading to diminished side-effects.

The use of the USA-Food and Drug Administration (FDA) and European Medicines Agency (EMA)-approved drugs for diseases for which they were not designed is known as drug repositioning [1] or repurposing. Celecoxib (CXB) is a nonsteroidal anti-inflammatory drug (NSAID) prescribed for treating rheumatic disorders through preferentially blocking cyclooxygenase type 2 (COX-2) activity (inhibition constant, $Ki$ = 10 μM) [2, 3]. In addition to its canonical anti-inflammatory activity, single-CXB treatment also inhibits colorectal (Caco-2, SW-480, and HT-29) and breast cancer cell (MCF-7, MDA-MB-231) growth at micromolar doses (10–50 μM) [4–7].

Although no apparent effect on non-cancerous cells (human breast epithelial MCF-10A cells, rat 3T3 fibroblasts) has been detected when using single-CXB treatment [6, 7], it is well-documented that CXB extensive use in the clinic is associated to multiple side effects [8–10]. Nevertheless, the use of combination therapy based on CXB, at adjusted lower doses, may represent a potentially promising approach for translating basic bench research into affordable clinical use.

In a previous study, CXB and 2,5-dimethylcelecoxib (DMC), an analogue of CXB with no apparent effect on COX-2 [11], was combined with two canonical chemotherapy drugs, paclitaxel (PA) or cisplatin (CP), to deter growth of human cervical HeLa, human cervix cancer SiHa and human glioblastoma U373 cancer cells in monolayer and HeLa multicellular tumor spheroids [12]. By using both Bliss independence model and resistance index assessments

[12], it was demonstrated that low CXB or DMC doses were able to induce a potent inhibitory synergistic effect when combined with PA and CP, decreasing proliferation (>80%) of low metastatic (SiHa) and high metastatic (HeLa and U373) cells, and spheroids. Preliminary results demonstrated that CXB and DMC combinations with PA or CP blocked OxPhos flux (>80%) and consequently cellular invasiveness was significantly decreased [12].

As the mechanisms involved in the drug combination action were not fully elucidated, and considering that in toxicological studies performed with repurposing drugs, it is common practice to carry out a thorough cell function analysis for the identification of the additional biochemical mechanisms triggered by the tested drugs [13], then a further and novel analysis of several different essential functions (i.e., energy metabolism, ATP-dependent processes, ROS induction, mitophagy onset, apoptosis/necrosis and DNA stability) of human cervix HeLa and SiHa cells was undertaken in the present study. This approach may help to understand the interactions of CXB or DMC and its combinations with multiple targets to propose alternative treatments for cervix cancer.

## Materials and methods

### Drugs

Celecoxib (CXB), dimethyl-celecoxib (DMC), cisplatin (CP) and paclitaxel (PA) were obtained from Sigma-Aldrich Chemical Co. (St Louis, MO, USA). CXB, DMC and PA were dissolved in 70% ethanol/30% DMSO, whereas CP was dissolved in distilled water [12].

### Cancer cell line

Human HeLa and SiHa cervix cancer cells were obtained from the ATCC (American Type Culture Collection). Cell genotyping analyses, performed by the National Institute of Genomic Medicine (INMEGEN, México), showed that HeLa and SiHa cells shared 87.5–90% alleles (14–15 from 16) reported by the ATCC for their authentication.

Cancer and 3T3 fibroblast cells ($1 \times 10^6$ cells/mL) were grown in Petri dishes with 20 mL Dulbecco's Modified Eagle's Medium high glucose (DMEM, 25 mM glucose; Sigma-Aldrich) in the absence or presence of selected drugs for 24 h. Cells were incubated under 5% $CO_2$ and 95% air at 37°C [12]. The drug concentrations used for the present study, which represents synergistic sub $IC_{50}$ concentrations found in HeLa cells [12] were 5 μM for celecoxib (CXB), 15 μM for dimethyl-celecoxib (DMC), 2 μM for cisplatin (CP) and 15 μM for paclitaxel (PA), either when they were used alone or in combination, CXB/CP or DMC/CP or CXB/PA or DMC/PA. In all experimentation, non-treated cells (*i.e.*, control cells) were incubated with the drug vehicle solution (70% ethanol/30% DMSO). The DMSO and ethanol concentrations added to the cells were lower than 10% which do not affect cell proliferation [6, 14].

### Cancer cell proliferation/viability assay

Cancer cells ($20 \times 10^3$ cells/well) were grown in 96-well plates containing DMEM (Sigma-Aldrich) for 24 h. Afterwards, cells were incubated with drugs alone or combinations at doses previously indicated for 24 h. The effect of these inhibitors on proliferation was determined on attached cells by using the 3-(4,5-dimethyl-thiazol-2-yl)-2,5-diphenyltetrazolium bromide (MTT) assay (Sigma-Aldrich). The agents used for the MTT assay did not affect the viability of any of the assayed cancer cells (viability was >95%) [12]. Cell protein was determined by Lowry [15] and Biuret [16] assays using bovine serum albumin (BSA) as standard. Viability assessed by the trypan blue exclusion assay revealed less than 10–15% cellular death under all experimental conditions for both tumor and normal cells [17].

## Western blot assay

Cancer cells were mixed with RIPA (PBS 1X, pH 7.2, 1% IGEPAL, 0.1% SDS and 0.05% sodium deoxycholate) lysis buffer *plus* 1 mM PMSF (phenyl methanesulfonyl fluoride) and 1 tablet of complete protease inhibitors cocktail (Roche, Mannheim, Germany), which was dissolved in 10 mL RIPA buffer. Once the protein concentration was determined by the Lowry method [15], the samples (50 μg) were re-suspended in loading buffer containing 10% glycerol, 2% SDS, 0.5 M Tris-HCl, 0.002% bromophenol blue, pH 6.8 *plus* 5% β-mercaptoethanol and loaded onto 10 or 12.5% polyacrylamide gel. Electrophoretic transfer to PVDF membranes (BioRad; Hercules, CA, USA) was followed by overnight immunoblotting at 4°C with 2OGDH, α-tubulin, actin, ATG-7, BAX, BCL-2, BID, BNIP-3, COX-IV, DRAM, GLUT 1, HKI, HKII, Ki67, LAMP-1, LDH, ND1, nucleolin, p21, PARK-2 (Parkin), PCNA, P-glycoprotein, PINK-1, SDH and XIAP antibodies at 1:500 dilution; and CASP-1 and CASP-3 antibodies at 1:1000 dilution (Santa Cruz; Santa Cruz, CA, USA). The hybridization bands were revealed with the corresponding secondary antibodies conjugated with horseradish peroxidase (Santa Cruz Biotechnology). The signal was detected by chemiluminescence using the ECL-Plus detection system (Amersham Bioscience; Little Chalfont, Buckinghamshire, UK). Densitometry analysis was performed using the Scion Image Software (Scion; Bethesda MD, USA) and normalized against its respective load control. Western blotting of each protein shown represents the mean ± S.D. of at least three independent experiments [17].

## Glycolytic and OxPhos fluxes and mitochondrial electrical membrane potential (ΔΨm) in cancer cells exposed to drug combinations

For glycolytic flux, cancer cells (1–3 mg protein/mL), cultured for 24 h with the indicated drugs, were harvested and further incubated in Krebs-Ringer (KR) buffer (125 mM NaCl, 5 mM KCl, 25 mM Hepes, 1 mM MgCl2, 1 mM KH2PO4, 1.4 mM CaCl2, pH 7.4) at 37°C under smooth orbital shaking as previously reported [18]. Glycolysis was started by adding 5 mM glucose (Sigma-Aldrich). Cellular samples were collected after 0 and 10 min of incubation, swiftly mixed with 3% (w/v) of cold perchloric acid (final concentration) and centrifuged at 3500 rpm for 5 min. Supernatants were neutralized with 1N KOH/100 mM Tris. Cells were also incubated with 2-deoxyglucose (2-DG, 20 mM) (Sigma-Aldrich) to prevent lactate production by glycogen degradation and external glucose-driven glycolysis. Lactate production was determined by the lactate dehydrogenase (Roche, Mannheim, Germany) coupled assay registering the NADH formation at 340 nm [19].

For total oxygen consumption and OxPhos fluxes, cancer cells (1 mg protein/mL), cultured for 24 h with the indicated drugs, were harvested and further incubated at 37°C in air-saturated KR buffer under constant agitation. To discard non-mitochondrial oxygen consumption, cells were incubated with 5 μM oligomycin (Sigma-Aldrich), a specific and permeable inhibitor of the mitochondrial ATP synthase. The OxPhos flux (*i.e.*, the rate of oligomycin-sensitive oxygen consumption) was determined by using a Clark type electrode, as described elsewhere [20], and a high-resolution respirometer (Oroboros Instruments, Innsbruck, Austria).

The contribution of glycolysis and OxPhos to the cellular ATP supply was calculated [21], respectively, from the rate of 2-DG-sensitive lactate production, assuming a stoichiometry of 1 mol of ATP produced *per* 1 mol of lactate generated and from the oligomycin-sensitive respiration rate multiplied by the ATP/O ratio of 2.5 (or the ATP/$O_2$ ratio of 5), which was determined in isolated cancer mitochondria [22].

The ΔΨm was determined by following the fluorescence signal of 0.25 μM rhodamine 6G incubated with control and drugs-treated HeLa cells (0.25 mg cellular protein/mL) in KR buffer at 37°C in a Shimadzu spectrofluorometer RF-5301PC (Tokyo, Japan); the excitation

and emission wavelengths were 480 and 565 nm, respectively. The magnitude of the fluorescence signal derived from mitochondria (*i.e.*, $\Delta\Psi$m) was established with the addition of the uncoupler CCCP (5 μM) [23].

## Mitophagy assay in cancer cells exposed to drug combinations

Mitophagy (selective removal of damaged mitochondria by autophagosomes and lysosomes) was detected by epifluorescence microscopy using an EVOS cell imaging microscope (ThermoFisher, Waltham MA, USA). Cancer cells (50 x $10^3$ cells/mL) were cultured in 2 mL DMEM medium in glass bottom culture dishes and exposed to drugs for 24 h. Afterwards, cells were loaded with 400 nM Bis-Benzimide H 33342 trihydrochloride (Hoechst), 500 nM MitoTracker Green FM and 500 nM LysoTracker Red for 30 min at 37°C. Cells were washed with DMEM medium without phenol red and analyzed in the EVOS microscope.

## ROS Determination in cancer cells exposed to drug combinations

Cancer cells (1 x $10^4$ cells/ml) were cultured in 96-well plates in DMEM in the presence of drugs for 24 h. After washing, cells were incubated with fresh DMEM in the absence of serum *plus* 25 μM dihydroethidium (DHE), and the fluorescent signal was monitored every 30 s for 1 h with a Varioskan microplate reader (Thermo Fisher Scientific; Waltham, MA, USA) using $\lambda_{emission}$ of 605 nm and $\lambda_{excitation}$ of 518 nm [24]. For ROS intracellular content, SiHa cells (1 x $10^4$ cells/ml) were cultured in 96-well plates in DMEM with or without drugs for 24 h. Afterwards, cells were washed and incubated with fresh DMEM in the absence of serum *plus* 25 μM dihydroethidium (DHE) for 1 h at 37°C. Afterwards, the DHE-fluorescence was monitored with a BioTeck cytation 7 microplate reader (Agilent Instruments, Santa Clara, CA, USA) using $\lambda_{emission}$ of 605 nm and $\lambda_{excitation}$ of 518 nm [25].

## Apoptosis and cell cycle distribution assays

To assess apoptosis, cells were cultured in the absence and presence of different drugs for 24 h. Afterwards, medium with non-attached cells was centrifuged at 2200 rpm for 3 min and cells were recovered and resuspended in phosphate buffer (PBS). Attached-cells were trypsinized and centrifuged at 2200 rpm for 3 min. Both non-attached and attached cells were mixed and centrifuged at 2200 rpm for 3 min. Afterwards, cellular sediment was suspended in 400 μL of PBS for FACS analysis. Cellular apoptosis was monitored with 25 ng Annexin V–FITC *plus* 250 ng propidium iodide in non-treated and drug-treated cancer cells and analyzed using a FACS Calibur flow cytometer (Becton Dickinson, San Jose, California, USA). Analysis was carried out for 10,000 events (1–2 min) using the BD CellQuest Pro Software (Becton Dickinson, NY, USA) [26].

For cell cycle distribution, cells were cultured in the absence and presence of different drugs for 24 h. Afterwards, medium with non-attached cells was centrifuged at 2200 rpm for 3 min and cells were recovered. Attached-cells were trypsinized and centrifuged at 2200 rpm for 3 min. Both non-attached and attached cells were mixed and centrifuged at 2200 rpm for 3 min. Afterwards, cells were washed with PBS and resuspended in 50% methanol buffer at 4°C and incubated for at least 10 min. Thereafter, cells were washed once with fresh PBS and incubated for 60 min with 50 U RNAse/mL at 37°C. Then, cells were washed with fresh PBS and incubated with 20 μL propidium iodide/mL for 2 min. The cell cycle distribution was analyzed with a flow cytometer FACS Calibur flow cytometer (Becton Dickinson, San Jose, California, USA). Analysis was carried out for 10,000 events (1–2 min) using the BD CellQuest Pro Software (Becton Dickinson, NY, USA) [27].

## P-glycoprotein activity

Cancer cells ($5 \times 10^4$ cells/ml) were seeded in 0.1 mL of DMEM in the presence of drugs. After 24 h, the remaining cells were loaded with 0.25 μM calcein-AM (Trevigen Inc., Helgerman, USA). Dye uptake was measured after 30 min incubation at 37˚C using $\lambda_{emission}$ of 517 nm and $\lambda_{excitation}$ of 494 nm [28].

## Cyclooxygenase (COX) activity assay

COX activity was performed by monitoring the rate of $O_2$ uptake using an oxymeter (YSI oxymeter model 5300A-1), which was coupled to an $O_2$ Clark type electrode. HeLa cell homogenate (100 μg protein) was added to initiate the reaction into a buffer containing 0.1 M Tris-HCl, 1 mM phenol, 85 μg bovine hemoglobin, pH 8 and 100 μM arachidonic acid (AA) at 37˚C. A unit of cyclooxygenase activity was defined as the ability of the enzyme to catalyze oxygenation of 1 nmol AA per minute at 37˚C. The calibration curve was made with human COX-2 (C0858-1000UN, Sigma-Aldrich) [29].

## DNA damage assay

For damaged DNA detection in HeLa cells, the Gentra Puregene kit (QUIAGEN, Hilden, Germany) was used as described by the manufacturer's instructions. DNA extraction was carried out by adding lysis solution (3 mL) to HeLa cells and incubating at room temperature for 5 min. Afterwards, cells were centrifuged at 2000 rpm for 5 min; the supernatant was recovered and incubated for 2 h at 37˚C. For DNA purification, sample was incubated for 15 min with RNAse A, protein precipitation solution and isopropanol and centrifuged at 10,000 rpm for 3 min. Sample was incubated with 70% (w/v) ethanol and centrifuged once for 1 min at 2,200 rpm. For DNA dissolution, sample was incubated with DNA hydration solution for 1 h and placed in 1% agarose gel. Afterwards, gel was immersed in 0.5X TBE (45 mM Tris, 45 mM boric acid; 1.2 mM EDTA, pH 8.0) buffer and electrophoresis was performed at 8V/cm for 30 min. Images were revealed with an UV transilluminator.

## Data analysis

Experiments were performed at least with three independent cell preparations (n). Data shown represent mean ± standard deviation (S.D.). ANOVA/post hoc Scheffé analysis was used with P values $< 0.05$ or lower to determine statistical significance [30].

# Results

## Effect of CXB/CP or CXB/PA combination on cancer cells and 3T3 fibroblasts proliferation

The present study was performed in two human cervical cancer cell lines, HeLa and SiHa cells, which represent two metastatic and drug resistant models in the field [31–33]. Our research group [12] recently demonstrated that subIC$_{50}$ concentrations of CXB (5 μM)/CP (2 μM) or CXB (5 μM)/PA (15 μM) and DMC (15 μM)/CP (5 μM) or DMC (15 μM)/PA (20 μM) showed a synergistic effect inhibiting the growth of HeLa cells after 24 h. Other drug combinations (at least 20 combinations for each drug) did not show significant synergism, or even showed an infra-additive effect on cancer cell growth [12].

Therefore, in the present study the same synergistic combinations of CXB (5 μM) /CP (2 μM) and CXB (5 μM) /PA (15 μM) were assayed on HeLa as well as on SiHa cells in order to validate and extend previous results on cell proliferation and mitochondrial function [12] and

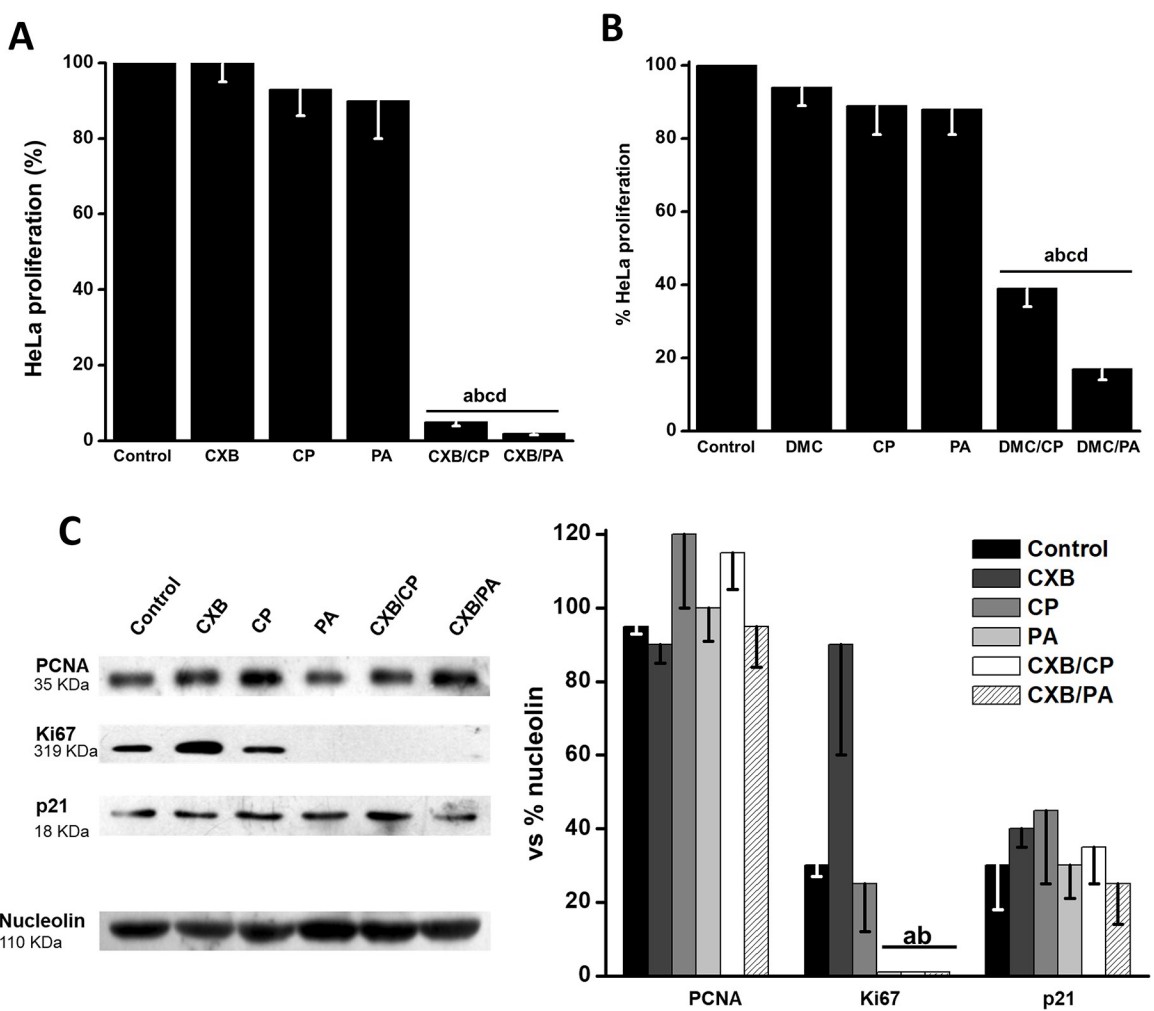

**Fig 1.** Effect of CXB and DMC combinations on HeLa (A), SiHa (B) and T3T (C) cell growth. The indicated drugs were added at the following concentrations: CXB (5 μM); CXB (5 μM)/CP (2 μM); CXB (5 μM)/PA (15 μM); DMC (15 μM), DMC (15 μM)/CP (5 μM) or DMC (15 μM)/PA (20 μM). Cells were exposed to the drugs for 24 h. Data shown represent the mean ± S.D. of at least three different preparations. *p < 0.05 *vs.* control (non-treated cells); **p < 0.05 *vs.* CXB or DMC.

to elucidate the main biochemical mechanisms involved. Drug combinations assayed rendered a marked decrease in the HeLa (>95%, Fig 1A) and SiHa (>75%, Fig 1B) cell proliferation *vs.* the single drug incubation. Similar results were obtained with DMC (15 μM) /CP (2 μM) and DMC (15 μM) /PA (15 μM) (Fig 1A and 1B). Both drug combinations did not induce any apparent inhibitory effect on non-cancer rodent 3T3 fibroblasts growth (Fig 1C).

To further support the above findings, cancer cell cycle distribution analyzing changes in Sub-G1, G0-G1, S and G2/M cell cycle phases induced by tested drugs, was performed in both cancer cells (S1A and S1B Fig). Combination treatment decreased both S and G2/M phases (CXB/PA, DMC/PA and DMC/PA) in HeLa cells, whereas proportion of cells in G0/G1 phase remained unchanged. Similar results were observed in SiHa cells (S1B Fig). These results indicated that CXB/PA, DMC/CP, and DMC/PA combinations inhibit DNA replication without affecting cellular mitosis.

These last results correlated with the low level of the proliferation marker Ki67 protein in HeLa cells incubated with CXB/CP or CXB/PA (S2 Fig). However, the single addition of PA

also promoted the lowering of Ki67 (6 times *vs.* control), which did not correlate with its null effect on cell growth. Perhaps, this result may be related with a non-canonical (moonlighting) function of Ki67, that of organizing heterochromatin compaction, rather than regulating proliferation [34]. In this regard, it has been shown that very low PA concentrations (0.1 μM/48 h) induce chromatin fusion and instability in mouse embryonic fibroblasts (MEFs), without affecting cell growth [35]. Thus, low Ki67 levels in PA-treated cells could be the result of chromatin disorganization. No changes were observed in p21, a protein related with the cell cycle inhibition through the cyclin kinase pathway or the proliferating nuclear antigen (PCNA) involved in DNA replication and repair (S2 Fig).

### Effect of CXB and DMC drug combinations on cancer energy metabolism

To test the hypothesis that energy metabolism is targeted by the NSAIDs combinations, the glycolytic and OxPhos protein levels in HeLa cells (S3 Fig), and pathway fluxes in HeLa (Fig 2A) and SiHa (Fig 2B) cells were determined after 24 h incubation with drug combinations at sub IC$_{50}$ concentrations. Total ATP supply derived from both glycolysis and OxPhos fluxes was also estimated in the presence and in the absence of tested drugs (Fig 3A and 3B).

The HKI level in HeLa cells was greater with CXB and similar with CXB combinations, in comparison to control (S3 Fig). Likewise, the HKII level was also greater with CXB alone, but it completely disappeared with CXB combinations. PA alone decreased the HK-II level by 63%. In contrast, the GLUT-1 and LDH-A content levels were not modified by drugs. The lower content in HK-II (low glucose affinity isoform) with CXB, PA or CP alone, and their combinations, apparently led to a significant glycolytic flux inhibition of 35–50% and 76–85%,

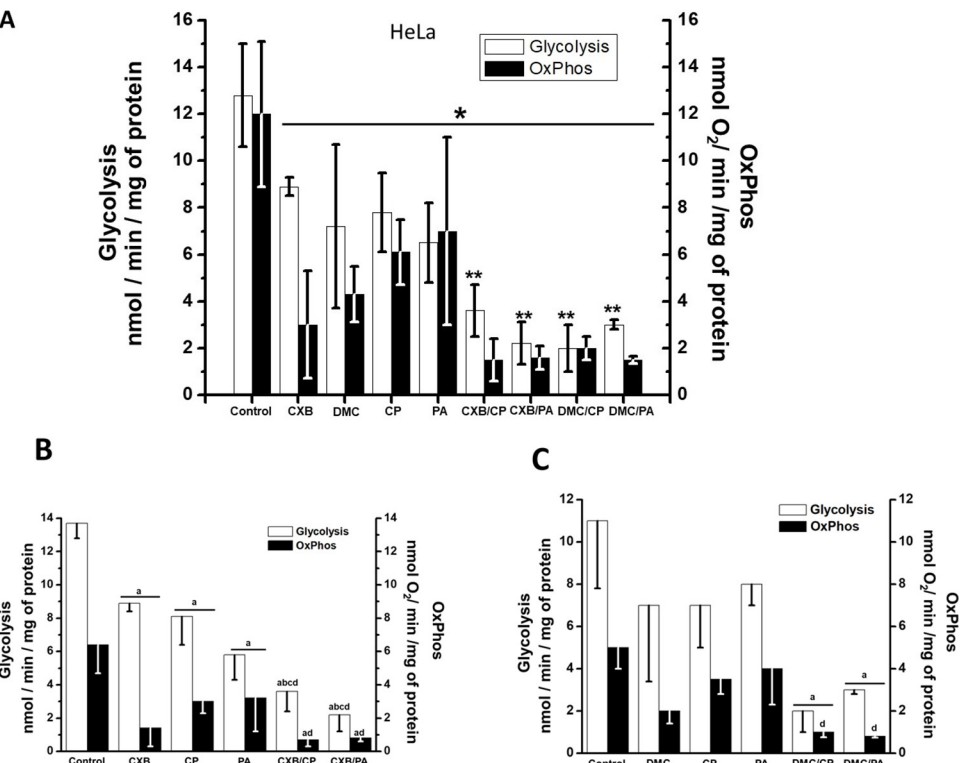

**Fig 2.** Effect of CXB and DMC combinations on HeLa (A) and SiHa (B) energy metabolism fluxes. For A, data shown represent the mean ± S.D. of 7 different preparations. For B, data shown represent the mean ± S.D. of at least three different preparations. *p < 0.05 vs. control (non-treated cells);**p < 0.05 *vs.* CXB or DMC.

respectively (Fig 2A). In turn, glycolysis in SiHa cells was similarly inhibited by CXB/PA and CXB/CP (Fig 2B).

CXB/PA and CXB/CP treatment in HeLa cells decreased the Krebs cycle 2-OGDH (90%) and SDH (50–75%) as well as the respiratory chain ND-1 (86%) and COX-IV (44–80%) enzyme levels (S3 Fig). These changes in protein contents after 24 h drug exposure promoted an abolishment of OxPhos flux near to 90% (Fig 2A). Moreover, 2-OGDH and SDH also significantly decreased (60–90%) by the addition of single CXB sub $IC_{50}$ doses (5 μM), which in turn promoted a drastic lowering in OxPhos flux. However, ND1 and COX-IV protein contents were not affected by single CXB. Although PA or CP alone significantly affected OxPhos enzyme contents (S3 Fig), the maximal inhibitory effect on the mitochondrial membrane potential (ΔΨm, 45–50%) (S4 Fig), OxPhos flux and ATP cell supply (87–88%) (Figs 2A and 3A) was attained with CXB/PA or CXB/CP, showing a potent synergistic effect of CXB when combined with PA or CP (Fig 3A).

The lack of linear correlation between OxPhos inhibition and ΔΨm decrease induced by the drug combinations (80% *vs*. 50%, respectively) is expected, since ΔΨm is a thermodynamic parameter derived from the logarithmic ratio of the $H^+$ concentrations at both sides of the inner mitochondrial membrane. In addition, a threshold ΔΨm of around -80 mV is required to drive ATP synthesis and compensatory mechanisms robustly maintain ΔΨm homeostasis [36, 37]. Comparable outcomes have been noted in HepG2 permeabilized cells after oligomycin treatment [38, 39].

Similar results on energy metabolism were attained in HeLa cells treated with DMC and its combinations (Figs 2A and 3A). In SiHa cells, combination of CXB or DMC with PA or CP also promoted an OxPhos flux and ATP cell supply inhibition higher than 70% (Figs 2B and 3B).

## Effect of CXB and DMC drug combinations on cancer mitochondrial digestion and ROS production

It has been documented that several NSAIDs activate mitophagy (mitochondrial degradation) in cancer cells [40]. Therefore, to assess the hypothesis that the cytotoxic effect of CXB and

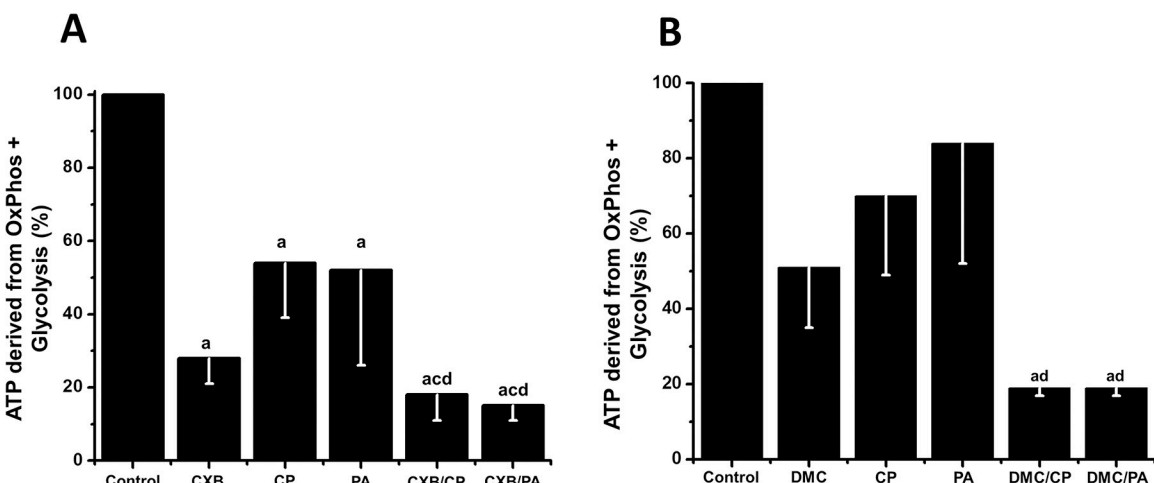

**Fig 3. Effect of CXB and DMC combinations on ATP supply derived from both glycolysis *plus* OxPhos in HeLa (A) and SiHA (B) cells.** For A, data shown represent the mean ± S.D. of at least 7 different preparations for CXB and the 100% value represents 46 ± 9 nmol ATP/ min/ mg cellular protein; for DMC, data shown represent the mean ± S.D. of three different preparations and the 100% value represents 36 ± 5.5 nmol ATP/ min/ mg cellular protein. For B, data shown represent the mean ± S.D. of three different preparations and the 100% value represents 38.5 ± 3.5 nmol ATP/ min/ mg cellular protein *p < 0.05 *vs*. control (non-treated cells); **p < 0.05 *vs*. CXB or DMC.

DMC combinations on OxPhos of cancer cells is mediated by mitophagy onset, several autophagy key proteins (HeLa) and direct mitophagy digestion (HeLa and SiHa) were analyzed in control and drug exposed cells. Indeed, increased contents in LAMP1 (20–60 times), BNIP-3 (10 times), ATG-7 (1.5–3.5-fold) and PARK-2 (Parkin, 15–30 times) protein levels were observed for HeLa cells exposed to CXB, PA, CP alone and CXB or DMC combinations *vs.* non-treated HeLa cells (Fig 4). Other autophagy proteins such as DRAM decreased by 2 times

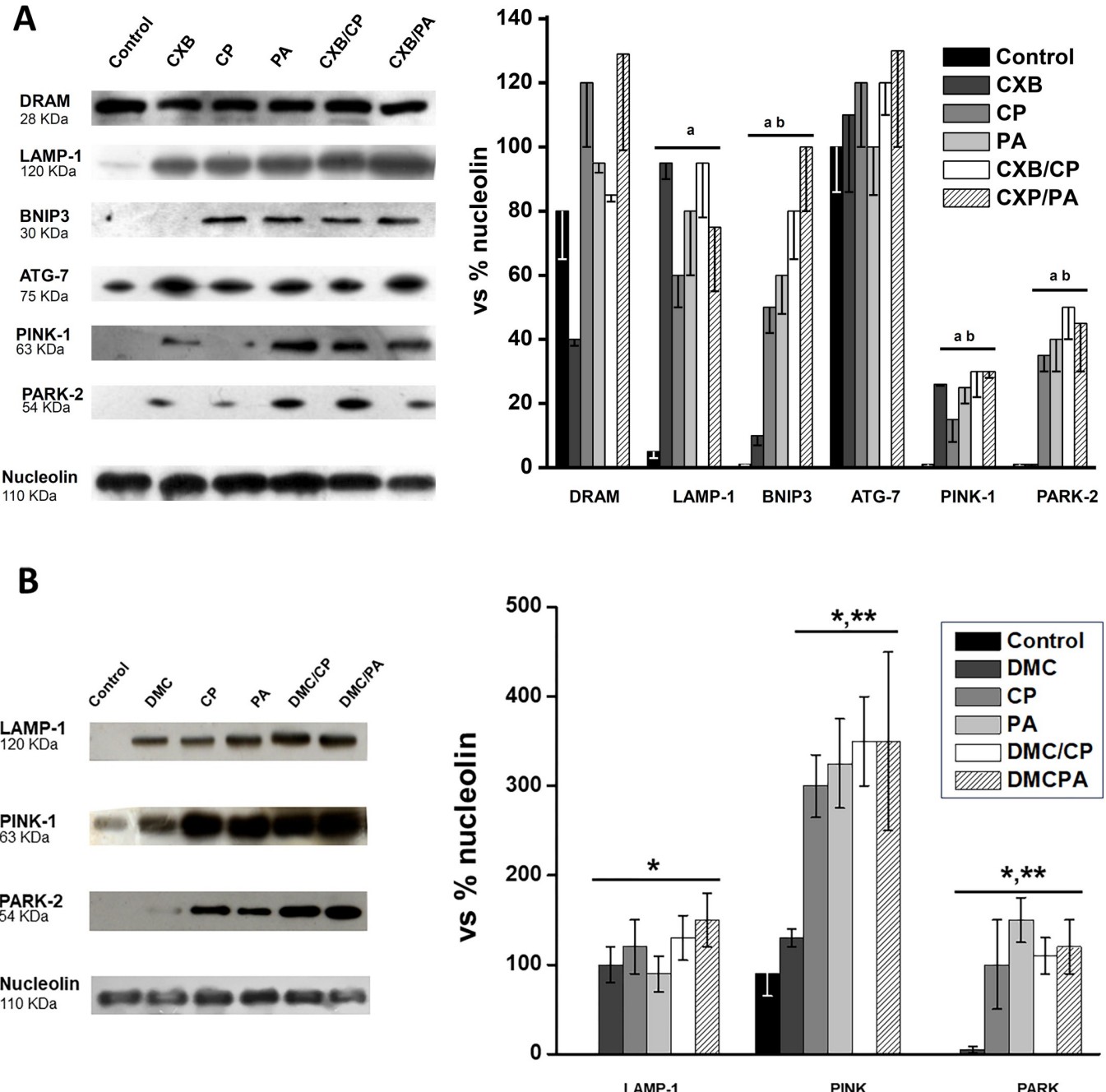

**Fig 4.** Effect of CXB (A) and DMC (B) combinations on the contents of proteins involved in mitophagy and mitochondrial fission. Data shown represent the mean ± S.D. of at least three different preparations. *p < 0.05 vs. control (non-treated cells); **p < 0.05 *vs.* CXB or DMC.

after CXB treatment. PINK-1, one of the mitochondrial proteins associated with mitochondrial dysfunction protection during cellular stress, significantly increased by 25–50 times in the presence of CXB or DMC and its combinations.

Increased autophagy protein contents in cells incubated with CXB or DMC combinations correlated with an increment in the content of autophagy vesicles for mitochondrial digestion in both HeLa (S5A, S5B Fig) and SiHa (S5C Fig) cells. Thus, in contrast to cells exposed to CXB, DMC, PA or CP alone, co-localization of mitochondria inside lysosomes, thus revealing mitophagy, was observed only in cells exposed to CXB or DMC combinations (S5 Fig).

Mitochondrial dysfunction is frequently associated with high ROS production [41]. Therefore, to explore whether oxidative stress was among the growth inhibitory mechanisms triggered by CXB and DMC combinations, ROS production in cancer cells was analyzed. Exposing HeLa cells for 24 h to subIC$_{50}$ CXB, DMC or CP concentrations brought about 1.7–2.1 times increase in ROS levels over the endogenous ROS baseline (Fig 5). No increased ROS production was obtained with PA alone. However, maximal ROS production was attained after CXB or DMC combination treatment (2.3–3.4 times) (Fig 5A). Likewise, the ROS levels in SiHa cells exposed to CXB or DMC combinations increased by 2.5–2.8 times (Fig 5B).

### Effect of CXB/CP and CXB/PA on cancer cell apoptosis and necrosis

To establish whether CXB/PA and CXB/PB induce apoptosis and necrosis to affect cancer cell growth, the apoptotic cell number (Fig 6A and 6B) as well as the levels of caspases 1 and 3 (Fig 6C) and apoptotic proteins (S6 Fig) was determined. As judged by the low content of HeLa and SiHa cells in early and late apoptosis (<5%) or necrosis (<4%) in flow cytometry (FC) analysis, it can be concluded that cancer cell death was not promoted by apoptotic or necrotic processes (S6 Fig). FC data revealed that cell viability was higher than 90% for all drug treatments (S6 Fig). The last results correlated with the negligible contents of caspase-1 and caspase-3 found in HeLa cells, supporting the observation that apoptosis was not triggered by CXB or DMC treatment.

For SiHa cells, caspase-3 was not present but caspase-1 levels increased by drugs, suggesting apoptotic induction. Further experimentation in SiHa cells must be performed in order to

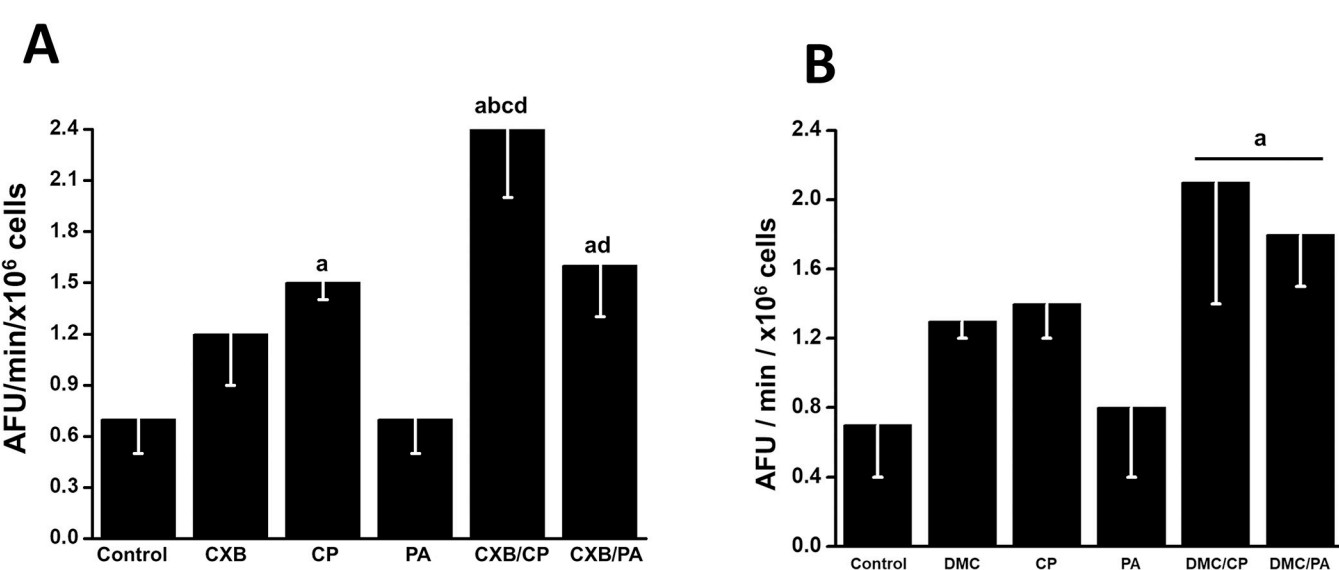

**Fig 5.** Effect of CXB and DMC combinations on ROS production (A) in HeLa and ROS content (B) in SiHa cells. Data shown represent the mean ± S.D. of at least three different preparations. *p < 0.05 *vs.* control (non-treated cells); **p < 0.05 *vs.* CXB or DMC.

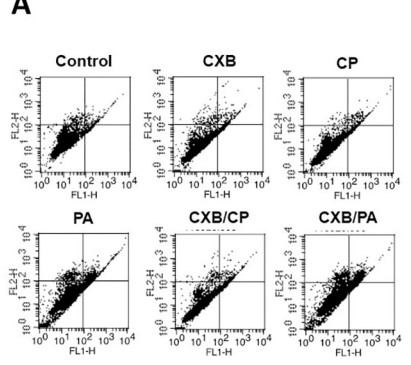

**A**

| | HeLa cells (%) | | | |
|---|---|---|---|---|
| | Alive | Early apoptosis | Late apoptosis | Necrosis |
| Control | 96.5 ± 3 | 0.8 ± 0.1 | 2 ± 0.2 | 0.8 ± 0.6 |
| CXB | 97 ± 2.7 | 1 ± 0.2 | 2 ± 1.2 | 0.5 ± 0.4 |
| CP | 91.5 ± 9.5 | 4 ± 0.4* | 4 ± 0.5* | 0.6 ±0.4 |
| PA | 94 ± 6 | 2 ± 0.3* | 3 ± 2 | 3 ± 0.2* |
| CXB/CP | 94 ± 2 | 2 ± 0.2* | 2 ± 1 | 2 ± 1* |
| CXB/PA | 92 ± 9 | 3.5 ± 0.5* | 3 ± 2 | 1 ± 0.2 |
| DMC/CP | 96 ± 1 | 0.3 ± 0.1 | 2 ± 0.6 | 0.6 ± 0.1 |
| DMA/PA | 97 ± 0.3 | 1 ± 0.2 | 2 ± 0.3 | 0.3 ± 0.1 |

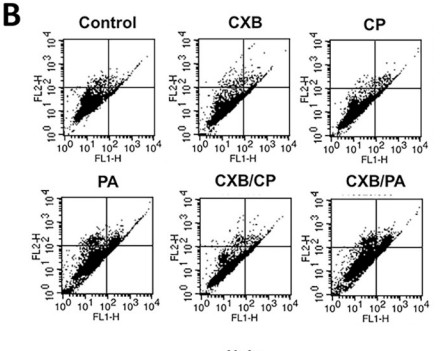

**B**

| | HeLa cells (%) | | | |
|---|---|---|---|---|
| | Alive | Early apoptosis | Late apoptosis | Necrosis |
| Control | 96.5 ± 3 | 0.8 ± 0.1 | 2 ± 0.2 | 0.8 ± 0.6 |
| CXB | 97 ± 2.7 | 1 ± 0.2 | 2 ± 1.2 | 0.5 ± 0.4 |
| CP | 91.5 ± 9.5 | 4 ± 0.4 | 4 ± 0.5 | 0.6 ±0.4 |
| PA | 94 ± 6 | 2 ± 0.3 | 3 ± 2 | 3 ± 0.2 |
| CXB/CP | 94 ± 2 | 2 ± 0.2 | 2 ± 1 | 2 ± 1 |
| CXB/PA | 92 ± 9 | 3.5 ± 0.5 | 3 ± 2 | 1 ± 0.2 |

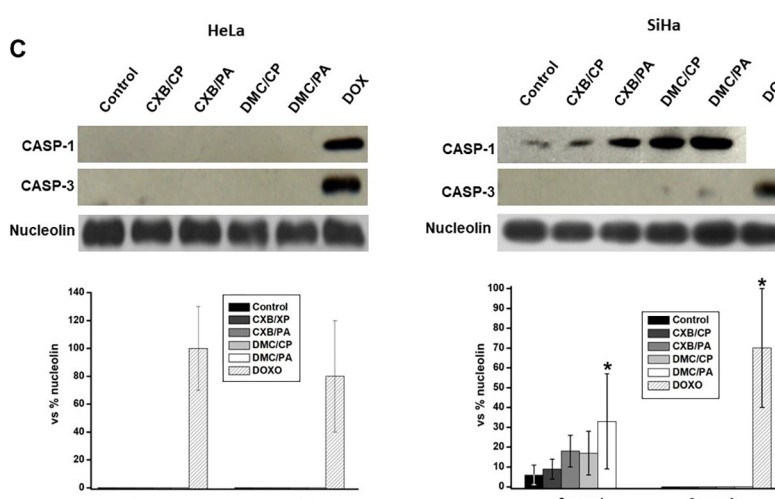

**Fig 6. Effect of CXB and DMC combinations on apoptosis in HeLa and SiHa cells.** Cell number (%) under apoptosis, determined by flux cytometry in HeLa (A) and SiHa (B) cells. (C) Contents of Caspase-1 and caspase-3 proteins. Data shown represent the mean ± S.D. of at least three different preparations. *p < 0.05 vs. control (non-treated cells). DOX, doxorubicin.

clearly demonstrate whether apoptosis is activated in this cancer line. As internal positive control for anti-caspases 1 and -3 detection, HeLa cells were exposed to 5 μM doxorubicin (DOX) for 3 h. As expected, DOX increased both apoptotic-induced caspases [42]. Pro-apoptotic protein contents (BAX and BID) were severely suppressed (58–99%) by the drug combinations in

HeLa cells compared to both non-treated cells and those incubated with a single drug. On the other hand, anti-apoptotic proteins (BCL-2, XIAP) were absent in non-treated cells but highly expressed (>30 times) in cells incubated with CXB/PA and CXB/CP (S6 Fig).

## Effect of CXB/CP and CXB/PA on P-glycoprotein content and activity

P-glycoprotein is an ATP-dependent plasma membrane transporter involved in the ejection of drugs from cytosol to the extracellular milieu in several cancer and non-cancer cells [43]. This protein is over-expressed in cancer cells. Hence, to examine whether this drug-elimination mechanism involved in CXB resistance was also affected by the drug combinations, the P-glycoprotein protein content and activity were assessed in HeLa (Fig 7A) and SiHa (Fig 7B) cells.

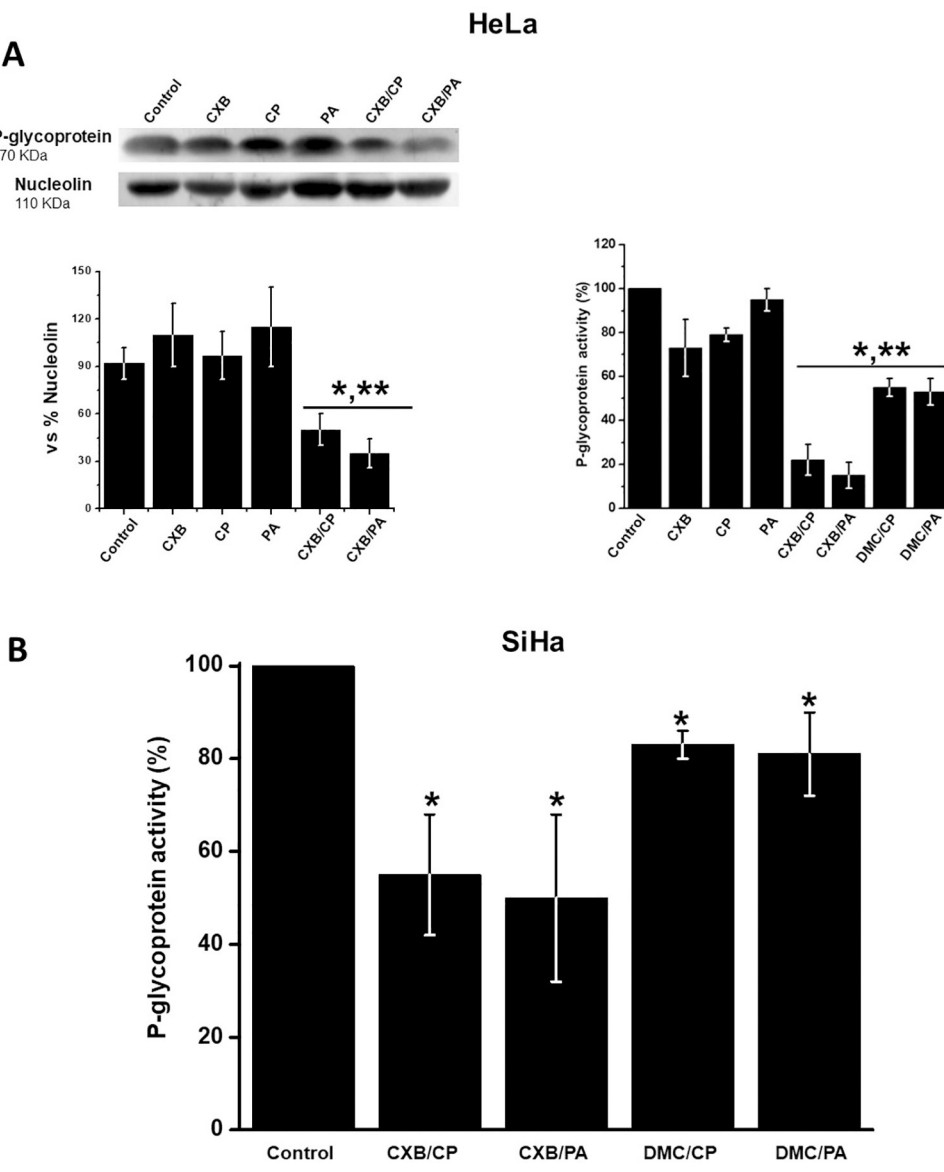

**Fig 7. Effect of CXB and DMC combinations on P-glycoprotein protein content and activity of P-glycoprotein in HeLa (A) and SiHa cells (B).** Data shown represent the mean ± S.D. of at least three different preparations. *$p < 0.05$ vs. control (non-treated cells); **$p < 0.05$ vs. CXB or DMC. The 100% activity value represents in (A) 2.5 ± 0.8 and in (B) 2 ±0.2 arbitrary units of fluorescence (AUF).

In HeLa cells incubated in the presence of single drugs, both P-glycoprotein content and activity were not significantly affected. However, CXB/PA or CXB/CP markedly decreased P-glycoprotein level (45–62%) and activity (48–85%). Similar results were observed in SiHa cells, where CXB/PA or CXB/CP decreased P-glycoprotein activity by 50–55%. HeLa and SiHa P-glycoprotein activity was less sensitive (20–50%) to DMC than CXB combinations.

### Effect of CXB/CP and CXB/PA on canonical targets

Finally, to examine whether the CXB combinations indeed affect their respective canonical targets, cyclooxygenase activity (CXB target), actin and α-tubulin levels (PA target) and DNA fragmentation (CP target) were also evaluated in HeLa cells (Fig 8).

The cyclooxygenase activity found in control HeLa cells was similar to that previously reported [44]. Addition of single low dose (5 μM) CXB or in combination, as well as CP and PA did not affect enzyme activity (Fig 8A), although a slight non-significant decrement was observed. As expected, 15 μM DMC did not significantly affect COX activity alone or in combination (Fig 8A). On the other hand, the content of α-tubulin was decreased solely in the presence of CXB/CP or CXB/PA (40–50%), whereas CXB, PA or CP alone showed a negligible effect. On the contrary, none of assayed drugs alone or in combination apparently affected the actin content (Fig 8B) and DNA stability (Fig 8C).

## Discussion

Drug repurposing is an emerging approach for treating diseases, in which pharmaceutical agents primarily used for non-cancerous diseases are being used for cancer treatment [45]. There are several examples of drug re-positioning with apparent initial success on glaucoma, epilepsy, and altitude sickness such as acetazolamide (a carbonic anhydrase inhibitor), which has also been used for the treatment of cancer. Others drugs like curcumin or resveratrol have been used to target cancer overexpressed anti-oxidant system [45].

However, conventional monotherapy is usually untargeted, lacking specificity for malignant cells and hence causing cancer as well as healthy cells to be perturbed. Therefore, combination therapy (CT) may represent a rational approach to gain therapeutic advantage by selectively targeting malignant cells thus achieving a more effective outcome. In general, CT targets multiple pathways thus promoting that cancer cells be unable to quickly adapt and respond to the simultaneous multi-site attack [46]. In addition, the use of CT may bring about several benefits over conventional monotherapy (MT). For instance, (i) since under MT, the lengthy treatment with a single compound induces and facilitates cancer cells to recruit and activate alternative salvage/resistant pathways (metabolic plasticity), then CT may decrease drug resistance because cells are exposed to simultaneous inhibition of several targets; (ii) CT can be more selective against actively proliferating cells because of its ability to target multiple pathways involved in cell proliferation; (iii) CT carries on higher efficacy, since it may decrease the drug concentration used for its ability to combine synergistically with other anti-cancer drugs [47]. This also attenuates the possible side effects of chemotherapy on patients.

### CXB repurposing in combination with canonical chemotherapy drugs to block metastatic cervix cancer growth through multitarget inhibition

Previous studies together with the present study have revealed that CXB, a non-steroidal anti-inflammatory drug, has high anticancer potential against several metastatic cancer cells such as human cervix, human breast, melanoma, colon and osteosarcoma cancer cells [5–7, 12, 48, 49]. An immediate interpretation of such observations may be that CXB anticancer effect is related to its cyclooxygenase 2 (COX-2, $Ki_{CXB} = 10$ μM) inhibition [2, 3]. However, the

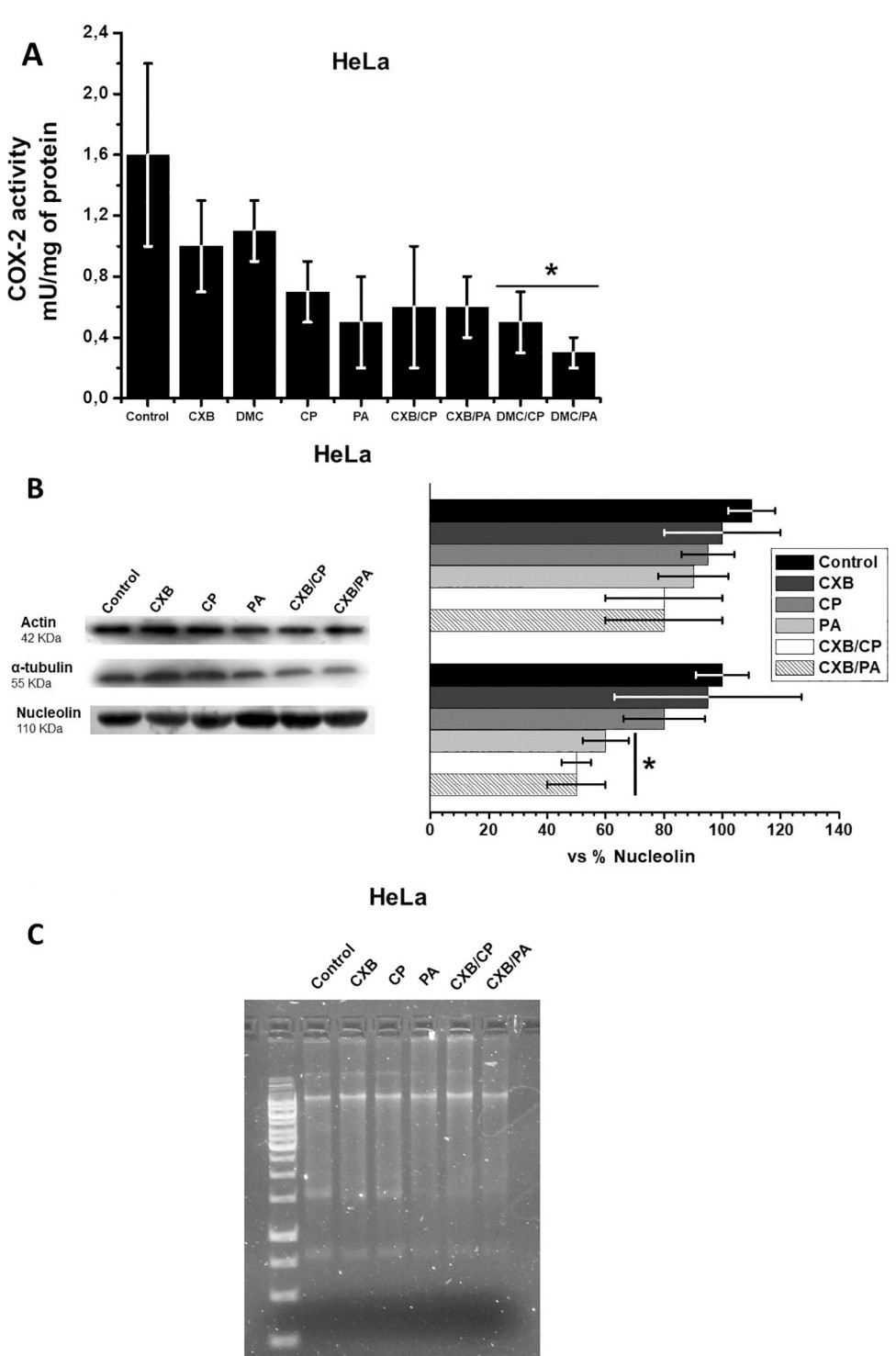

**Fig 8.** Effect of drug combinations on COX-2 activity (A), microtubules protein contents (B) and DNA integrity (C) in HeLa cells. Data shown represent the mean ± S.D. of at least three different preparations. *p < 0.05 vs. control (non-treated cells).

celecoxib analogue, 2,5-dimethylcelecoxib (DMC), which has no effect on COX-2 activity [11, 50], also exerts potent anti-cancer effects (*c.f.* Fig 1A and 1B) [12]. Furthermore, other essential cell functions such as OxPhos and OxPhos-dependent (or ATP-dependent) processes like metastasis [21] are also severely affected by CXB when it is used alone or in combination with PA or CP [12].

CXB/PA and CXB/CP target mitochondria by decreasing mitochondrial protein contents, OxPhos flux and mitochondrial transmembrane potential ($\triangle\psi$m). DMC also exhibits inhibitory effects on OxPhos and synergistic inhibition when combined with CP or PA (*c.f.* Figs 2 and 3) [12]. Similar effects have been observed in human breast cancer and murine melanoma cells exposed to higher CXB concentrations [7].

These drug combinations mainly affect mitochondria of cancer cells maybe because tumor plasma and inner mitochondrial membranes contain greater contents of cardiolipin (increasing the density of membrane negative charges) and cholesterol (decreasing the passive diffusion of protons across the membranes), facilitating the uptake and accumulation of lipophilic drugs [51]. Besides, solid tumor cells and their mitochondria are exposed to hypoxic microenvironments in which $O_2$ levels fluctuate temporally and spatially from normoxia (10%-21% atmospheric $O_2$) to hypoxia (0.1%-5% $O_2$). Hypoxia is related with an increment in intracellular ROS levels, which in turn has been associated with the development of metastatic phenotype. Therefore, the identification of pro-oxidant drugs (*i.e.*, NSAIDs) against solid tumors emerges as a promising and selective alternative for clinical treatment [52].

The lowering in the content of glycolytic and mitochondrial proteins induced by CXB/PA and CXB/CP may be related to the generalized inhibition of protein synthesis induced by CXB (30 μM/ 24 h), which is mediated by phosphorylation/inactivation of the eukaryotic translation initiation factor 2 alpha in HeLa cervix and PC-3 pancreatic cancer cell lines [53]. These inhibitory effects were not observed with other NSAIDs (rofecoxib, valdecoxib, indomethacin or flurbiprofen); except for DMC, suggesting that protein synthesis does not depend on COX-2 and that there are other sites of action for this NSAID. In this regard, it has been demonstrated that CXB does not affect human muscle protein synthesis [54]; although high doses (close to 100 μM) decrease protein synthesis in embryo fibroblasts [53]. Unfortunately, there is not available literature where CXB be used at lower concentrations or combined with other drugs related to protein synthesis.

Glycolysis was also affected by CXB/CP and CXB/PA. Scarce information has been published regarding the impact of CXB on glycolysis. However, CP (20 μM/48 h) decreases GLUT1 and LDH (>50%) protein contents as well as lactate production (60–70%) in human cervix (Siha) and human breast (MDA-MB-231) cancer cells through an integrin β5-mediated mechanism [55]. PA (25 μM/ 90 min) also affects glycolysis diminishing the levels of two major glycolytic activators fructose-1,6-biphosphate and glucose-1,6-bisphosphate, and causing detachment of PFK-1 from cytoskeleton in B16 F10 mouse melanoma cells [56]. However, both CP and PA concentrations used in these last studies were comparatively elevated and likely cause marked side effects on non-cancer cells [57]. Unfortunately, non-cancer cells were not used as control in those previous reports for comparison purposes and selectivity assessment.

Lowering in the ATP supply (>80%), derived from the energy pathways inhibition by CXB or DMC combinations did not cause cell death. Instead, cells maintained a metabolic quiescence in which resting ATP was used to maintain viability. Similar observations have been described in apyrase-treated cells where oxygen consumption and intracellular ATP decreased > 60%, but viability was no compromised [58]. It should be noted that ATP supply is a dynamic flux process related to metabolic rates, but it is not directly related to the cell steady-state ATP concentration, which most likely might not have been significantly altered.

To achieve a stable ATP homeostasis under lower ATP supply, it is then required to concomitantly lower the ATP demand.

## Effect of CXB and DMC combinations on ROS production and mitophagy activation

CXB and DMC promote cancer cell death by inhibiting mitochondrial functions and inducing increased superoxide production as a by-product from the electron transport chain [6, 7, reviewed in 52]. In the present study, single and subIC$_{50}$ NSAIDs concentration induced increased ROS production as result of both OxPhos (CXB/DMC) and $\Delta\psi$m (CXB) inhibition. It is well known that $\Delta\psi$m is the main component of the electrochemical proton (H$^+$) gradient across the inner mitochondrial membrane, which drives ATP production via ATP synthase [59]. However, the ROS production increases when the respiratory chain and ATP synthase become inhibited (by CXB or DMC and their combinations). This derives from a build-up in the levels of reduced intermediaries along the respiratory chain (FMNH$_2$, FADH$_2$, CoQH$_2$) [59, 60].

CXB/CP but not CXB/PA induced an enhanced ROS production (60% more *vs.* CXB alone). CXB/CP might be inducing massive ROS production by acting at additional sites other than OxPhos. In this regard, it has been found that low CXB doses (1.8 μM/24 h) in combination with glucosamine decreases the gene expression of several antioxidant enzymes such as SOD-2 and CAT, as well as NRF2, the transcriptional factor regulating redox cell status, in human osteoarthritic chondrocytes [61]. Unfortunately, antioxidant system protein contents and activities were not assessed to firmly conclude that CXB combination treatment affects cellular redox balance through this mechanism.

One mechanism involved in the low OxPhos activity found in HeLa cells exposed to CXB or DMC and their respective combinations was the mitochondrial fission activation observed by the overexpression of PINK-1 and PARK-2 proteins. Mitochondrial fission functions by segregating dysfunctional mitochondria for degradation through mitochondrial autophagy (mitophagy) [62], which in turn was observed by the significant increment in the autophagy proteins LAMP-1 and BNIP-3. Consequently, an active mitochondrial digestion was detected in both HeLa and SiHa cells (*cf.*, Fig 4; S5 Fig).

It has been documented that mitochondrial fragmentation has been associated with increments in oxidative stress, mitochondrial depolarization, and decreased ATP production [63]. Indeed, these three events were clearly attained in HeLa cells exposed to CXB drug combination treatment. In this regard, it has been recently demonstrated that most of the cytotoxic anticancer drugs currently used in the clinic induce mitophagy in cancer cells [64, 65].

## CXB drug combinations did not promote cellular death by apoptosis or necrosis

It is widely documented that CXB alone at high doses (20–100 μM) induces apoptosis in colon (HT-29), leukemia (HL-60), melanoma (B16F10) and breast (4T1) cancer cells [7, 66, 67]. In contrast, in cutaneous squamous cancer SCL-II cells, high CXB doses (50 μM/24 or 42 h) showed little effect on apoptosis and cell viability [68]. However, CXB (50 μM) combined with TRAIL or TRAIL-receptor agonists induced massive apoptosis in squamous cancer cells [68].

In the present study, low doses of CXB or DMC and their combinations with low doses of PA or CP induced a negligible early or late apoptosis in both assayed cancer cells, along with scarce presence of two important apoptosis markers, CASP-1 and CASP-3 [69] in HeLa cells and perhaps in SiHa cells. These observations suggest that subIC$_{50}$ CXB or DMC/CP/PA concentrations may not affect signalling pathways associated to apoptosis/necrosis activation.

Thus, it seems that the main cell death mechanisms associated with CXB/DMC and their combinations are mitophagy and mitochondrial fragmentation instead of apoptosis/necrosis. Cisplatin did not induce apoptosis in HeLa cells because of the low subIC$_{50}$ concentrations used in this study (*c.f.* Fig 6A). Similar results were found in human HL-60 cells treated with 1–3 μM CP for 24 h [70]. On the contrary, after CP treatment at 21 μM for 24 h [71, 72] or longer incubation times (96 h) [70], cancer cells undergo apoptosis activation [70].

After CXB and combination treatment, the anti-apoptotic proteins BCL-2 and XIAP significantly increased (*c.f.* S6 Fig), whereas pro-apoptotic proteins decreased as it has been documented in HeLa cervix and PC-3 pancreatic cancer cell lines after drug treatment because of severe protein synthesis inhibition [53]. There seems to be compensatory mechanisms to prevent HeLa apoptosis; thus BCL-2 and XIAP could not be considered as anti-apoptotic biomarkers after drug treatment. There are not comparative studies where this effect has been observed and further experimentation is required to clarify this observation.

## CXB/DMC combinations affect P-glycoprotein and cell microtubules but they do not induce DNA fragmentation or COX-2 inhibition

In cancer cells, the overexpression of P-glycoprotein, an ATP-binding cassette (ABC) transporter, is responsible for the ejection of several hydrophobic chemotherapeutic agents from cancer cells, causing low chemotherapy efficacy [43, 73]. P-glycoprotein is an ATP-dependent protein which hydrolyzes 2 ATPs to transport its substrates across the plasma membrane against a concentration gradient [74, 75]. Indeed, CXB/PA and CBX/CP decreased P-glycoprotein activity. Clearly, the synergistic action of CBX with the canonical anticancer drugs activate other mechanisms to affect cancer cell functions such as P-glycoprotein inhibition.

Regarding cell microtubules proteins, it has been reported that CXB alone (50–100 μM/48 h) does not induce degradation of β-actin [49] and α-tubulin [76] in several cancer cells. A similar result was obtained in the present study with HeLa cells. However, we observed a synergistic effect of the combined use of CXB and canonical chemotherapy drugs on microtubules protein contents. A somewhat different result was reported in rat injured spinal tissues treated with CXB *plus* fasudil (a rho-associated kinase inhibitor) [77] and in human non-small cell lung cancer treated with CXB *plus* aspirin [78], where the content of β-actin [77] and α-tubulin [78] was not affected by the drug combination.

Celecoxib is a well characterized inhibitor of prostaglandin-endoperoxide synthase or cyclooxygenase-2 (COX-2), which has been associated with metastatic phenotype maintenance [79, 80]. CXB alone at subIC$_{50}$ growth doses (5 μM) did not significantly affect COX-2 activity. The most likely reason for this result is that the CXB concentration used was insufficient to decrease COX-2 activity ($Ki_{celecoxib}$ = 10 μM) [2]. Moreover, DMC at low doses was also unable to affect COX-2 activity.

CP, PA, CXB/CP and CBX/PA exerted a slight but not significant inhibitory effect on COX-2 activity, because they were added at low doses. An explanation could be that CP and PA directly inhibit COX activity. Alternatively, the oxygen consumption methodology used for COX activity detection is not specific. Indeed, it has been described that the oxygen consumption methodology as well as others like the peroxidase co-substrate oxidation assay, radiolabeled chemical inhibition assay, and enzyme-linked immunosorbant assay (ELISA) for COX activity determination have limitations promoting low specificity and low reproducibility [81]. Thus, other methodologies such as liquid chromatography-tandem mass spectrometry (LC-MS-MS) have been developed for the rapid and accurate quantitative analysis of the COX product, prostaglandin-E2 [81]; however, there are not available studies with this last methodology indicating that PA or CP block COX activity.

None of drugs assayed promoted DNA degradation, including CP which binds to N7 reactive purine residues promoting DNA destabilization and in turn, cell death [82]. However, the concentration at which CP alone or in combination with CXB (100 μM), promotes DNA damage and other related processes like PI3K/Akt-associated anokis is 10 μM [49, 83]. Thus, the CP concentration used in this study (2 μM) seemed insufficient to significantly alter DNA integrity, but in combination with CXB becomes an effective anticancer drug.

Chemotherapy is still the gold standard for cancer treatment, although it is very often associated with severe adverse side effects and the development of drug resistance. The data of the present study in cervix cancer cells indicated that the drug combination of repurposed NSAIDs (CXB and DMC) *plus* canonical chemotherapy (PA or CP) represents a promising strategy,

i. to overcome the side effects related with the high drug concentrations currently used in monotherapy;

ii. to increase selectivity against cervix cancer cells for its ability to target multiple pathways such as energy (ATP) supply, ROS production, DNA stability, and drug defense; and

iii. to extend variations of the drug combinations, for instance with CXB/PA/CP or DMC/PA/CP at low doses on other cancer cell types and to be considered as potential treatment in the clinical practice and clinical trials. In this regard, preliminary observations indicated that SiHa cells (squamous carcinoma stage II) showed greater sensitivity to three-drug treatment than HeLa cells (adenocarcinoma stage IV), which warrants further experimentation to unveil which biochemical mechanisms are associated with this differential response.

## Supporting information

**S1 Fig.** Effect of CXB and DMC combinations on cell cycle phases in HeLa (A) and SiHa (B) cells. The indicated drugs were added at the following concentrations: CXB (5 μM); CXB (5 μM)/CP (2 μM); CXB (5 μM)/PA (15 μM); DMC (15 μM), DMC (15 μM)/CP (5 μM) or DMC (15 μM)/PA (20 μM). Cells were exposed to the drugs for 24 h. Data shown represent the mean ± S.D. of at least three different preparations. *$p < 0.05$ *vs.* control (non-treated cells).
(DOCX)

**S2 Fig. Proliferation marker protein contents in CXB treated-HeLa cells.** Data shown represent the mean ± S.D. of at least three different preparations. *$p < 0.05$ *vs.* control (non-treated cells); **$p < 0.05$ *vs.* CXB.
(DOCX)

**S3 Fig. Effect of CXB and DMC combinations on the contents of glycolysis and OxPhos proteins in HeLa cells.** Data shown represent the mean ± S.D. of at least three different preparations. *$p < 0.05$ *vs.* control (non-treated cells); **$p < 0.05$ *vs.* CXB or DMC.
(DOCX)

**S4 Fig. Effect of CXB combinations on the mitochondrial membrane potential (Δψm) in HeLa cells.** Data shown represent the mean ± S.D. of at least three different preparations. *$p < 0.05$ *vs.* control (non-treated cells). AUF, arbitrary units of fluorescence.
(DOCX)

**S5 Fig.** Epifluorescence images of HeLa (A, B) and SiHa cells (C) loaded with Mitotracker-green (MTG, 500 nM) and Lysotracker-red (LTR, 500 nM) in a complete DMEM medium after (A) CXB (n = 25), (B) DMC (n = 10) or (C) CXB/DMC (n = 10) treatment. Images were

taken with the EVOS FL (Thermo Fisher Scientifc Waltham, MA, USA) cell imaging micro-scope using a 60×objective. Bars = 50 μm. White arrows indicate the mitochondria/lysosome dye co-loading.
(DOCX)

**S6 Fig. Effect of CXB combinations on pro- and anti-apoptotic protein contents in HeLa cells.** Data shown represent the mean ± S.D. of at least three different preparations. *p < 0.05 *vs*. control (non-treated cells); **p < 0.05 *vs*. CXB.
(DOCX)

## Acknowledgments

This paper is part of the requirements for obtaining a Doctoral degree at the Posgrado en Ciencias Biológicas, UNAM of DXRC. The present work was partially supported by grants from CONAHCyT-México to DXRC (No. 814560); CONAHCyT-México (No. 283144) and PAPIIT, DGAPA-UNAM, México to SRE (No. IA201823), CONAHCyT-México (No. 6379) and National Institute of Chemical Physics and Biophysics (NICPB), Tallinn, Estonia Institutional Development Fund to RMS. DXRC and RLM thank CoreLab-INC for the flow cytometer use. There was no additional external funding received for this study. The funders had no role in study design, data collection and analysis, decision to publish, or preparation of the manuscript.
    Authors confirm that the minimal data set is present entirely within your manuscript.

## Author Contributions

**Conceptualization:** Sara Rodríguez-Enríquez.

**Data curation:** Diana Xochiquetzal Robledo-Cadena.

**Formal analysis:** Diana Xochiquetzal Robledo-Cadena, Silvia Cecilia Pacheco-Velázquez, Jorge Luis Vargas-Navarro, Joaquín Alberto Padilla-Flores, Rafael Moreno-Sánchez, Sara Rodríguez-Enríquez.

**Funding acquisition:** Rafael Moreno-Sánchez, Sara Rodríguez-Enríquez.

**Investigation:** Diana Xochiquetzal Robledo-Cadena, Silvia Cecilia Pacheco-Velázquez, Jorge Luis Vargas-Navarro, Joaquín Alberto Padilla-Flores, Rebeca López-Marure, Tuuli Kaambre, Rafael Moreno-Sánchez, Sara Rodríguez-Enríquez.

**Methodology:** Diana Xochiquetzal Robledo-Cadena, Silvia Cecilia Pacheco-Velázquez, Jorge Luis Vargas-Navarro, Joaquín Alberto Padilla-Flores, Rebeca López-Marure, Israel Pérez-Torres.

**Resources:** Sara Rodríguez-Enríquez.

**Supervision:** Sara Rodríguez-Enríquez.

**Validation:** Jorge Luis Vargas-Navarro, Rafael Moreno-Sánchez, Sara Rodríguez-Enríquez.

**Visualization:** Tuuli Kaambre.

**Writing – original draft:** Diana Xochiquetzal Robledo-Cadena, Rafael Moreno-Sánchez, Sara Rodríguez-Enríquez.

**Writing – review & editing:** Tuuli Kaambre, Rafael Moreno-Sánchez, Sara Rodríguez-Enríquez.

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
