## [Decision Letter · Decision Letter 0]

1 Mar 2024

PONE-D-24-03423Synergistic celecoxib and dimethyl-celecoxib combinations block cervix cancer growth through multiple mechanismsPLOS ONE

Dear Dr. Rodriguez-Enriquez,

Thank you for submitting your manuscript to PLOS ONE. After careful consideration, we feel that it has merit but does not fully meet PLOS ONE’s publication criteria as it currently stands. Therefore, we invite you to submit a revised version of the manuscript that addresses the points raised during the review process. Please submit your revised manuscript by Apr 15 2024 11:59PM.. If you will need more time than this to complete your revisions, please reply to this message or contact the journal office at plosone@plos.org. Please include the following items when submitting your revised manuscript:A rebuttal letter that responds to each point raised by the academic editor and reviewer(s). You should upload this letter as a separate file labeled 'Response to Reviewers'.A marked-up copy of your manuscript that highlights changes made to the original version. You should upload this as a separate file labeled 'Revised Manuscript with Track Changes'.An unmarked version of your revised paper without tracked changes. You should upload this as a separate file labeled 'Manuscript'.

We look forward to receiving your revised manuscript.

Kind regards,

Subhadip Mukhopadhyay, PhD

Academic Editor

PLOS ONE

Journal Requirements:

“The present work was partially supported by grants from CONAHCyT-México to DXRC (No. 464032), CONAHCyT-México (No. 283144) and PAPIIT, DGAPA-UNAM, México to SRE (No. IA201823), CONAHCyT-México (No. 6379) and National Institute of Chemical Physics and Biophysics (NICPB), Tallinn, Estonia Institutional Development Fund to RMS”

“This paper is part of the requirements for obtaining a Doctoral degree at the Posgrado en Ciencias Biológicas, UNAM of DXRC. The present work was partially supported by grants from CONAHCyT-México to DXRC (No. 464032); CONAHCyT-México (No. 283144) and PAPIIT, DGAPA-UNAM, México to SRE (No. IA201823), CONAHCyT-México (No. 6379) and National Institute of Chemical Physics and Biophysics (NICPB), Tallinn, Estonia Institutional Development Fund to RMS”

“The present work was partially supported by grants from CONAHCyT-México to DXRC (No. 464032), CONAHCyT-México (No. 283144) and PAPIIT, DGAPA-UNAM, México to SRE (No. IA201823), CONAHCyT-México (No. 6379) and National Institute of Chemical Physics and Biophysics (NICPB), Tallinn, Estonia Institutional Development Fund to RMS”

7. We note that you have included the phrase “data not shown” in your manuscript. Unfortunately, this does not meet our data sharing requirements. PLOS does not permit references to inaccessible data. We require that authors provide all relevant data within the paper, Supporting Information files, or in an acceptable, public repository. Please add a citation to support this phrase or upload the data that corresponds with these findings to a stable repository (such as Figshare or Dryad) and provide and URLs, DOIs, or accession numbers that may be used to access these data. Or, if the data are not a core part of the research being presented in your study, we ask that you remove the phrase that refers to these data.

Reviewers' comments:

Reviewer's Responses to Questions

**Comments to the Author**

1. Is the manuscript technically sound, and do the data support the conclusions?

Reviewer #1: No

Reviewer #2: Partly

2. Has the statistical analysis been performed appropriately and rigorously? 

Reviewer #1: Yes

Reviewer #2: No

3. Have the authors made all data underlying the findings in their manuscript fully available?

Reviewer #1: No

Reviewer #2: Yes

4. Is the manuscript presented in an intelligible fashion and written in standard English?

Reviewer #1: Yes

Reviewer #2: Yes

5. Review Comments to the Author

Reviewer #1: Robledo-Cadena et al has reported data suggesting synergistic association between celecoxib and dimethyl-celecoxib when used as a combinatorial therapy against cervical cancer by blocking its growth. Although, the study proposes exploratory avenues and scope of combinatorial therapy against cervical cancer, it is clear that further experimentation is required to fully validate the results. Multiple cell-lines as well alternative approaches should be utilized and the discussion should highlight the results pertinent to this study.

Line 55-57: This statement is broad. Combinatorial therapy and its advancement in the field of cancer research is already known. The authors need to focus on why this study or to be more specific the findings of this study is promising chemotherapy strategy against cervical cancer growth compared to what is already known in the literature.

The introduction needs work. What point are the authors trying to convey in the two paragraphs remains unclear. It is also confusing that Ralph, S.J.; et al. already reported CXB and DMC combinations with PA or CP blocked OxPhos flux (>80%) and consequently cellular invasiveness, so why did the authors choose to further analyze on several different functions of human cervix HeLa cells? What was the hint that additional biochemical pathways were involved? Could the authors explicitly clarify this point?

Important: All critical and key findings in this paper should be repeated in one additional cell lines like SiHa/C-33A to support the findings.

All bar graphs should show data points and the error bars should be + and – both.

Major points:

Fig1. A and B What happens when you add CXB/CP/PA or DMC/CP/PA? What is the control here? Non-treated or DMSO? Fig1. A and B can be merged.

Fig1. C All uncropped western images should be reported in the supple info.

Fig1. C the single addition of PA also promoted the lowering of Ki67 (6 times vs. control) in HeLa cells, which did not correlate with its null effect on cell growth, why?

Important: The expression levels should also be compared using additional techniques like qPCR.

Fig2. A Why is the level of HKI in CXB treated is significantly elevated compared to the control? Moreover, levels of CXB/CP treated looks very similar to controls? This is confusing and needs to be repeated and clarified. The HKII in CXB/CP and CXB/PA treated lanes seems completely diminished. What is this telling us? Is HKII a better marker compared to HKI? Overall, the western qualities are not very good. The Nucleolin loading standard almost seems over saturated. Lanes should be marled properly.

Although PA or CP alone significantly affected OxPhos enzyme contents (Fig 2A), the maximal inhibitory effect on the mitochondrial membrane potential (ΔΨm, 45-50%) (data not shown), Why?

Where is the western analysis for DMC/CP and DMC/PA for Glycolysis and OxPhos?

Fig3. A and B can be merged and should be presented with Fig2. B and C.

Fig4. BNIP-3 (10-100 times) increase? This range does not make sense and needs to be re-visited.

Fig6. These experiments should be repeated with the usual chemiluminescent assay apoptosis or cell proliferation methods and the results should be correlated with what is now Fig6. A. Where is the result for DMC/CP and DMC/PA in Fig. 6B?

Important: It is not clear to this reviewer as to why the effect of CXB/CP and CXB/PA on P-glycoprotein content and activity was measured? What was the rationale? Is it clear at this point the MOA of these drugs that the authors considered testing for resistance? Where is the result for DMC/CP and DMC/PA in Fig. 7?

Fig. 8 and 9 should be combined into one figure for better flow.

Not sure why the discussion and conclusion are separate? Please follow the submission guidelines.

Discussion: Line 467-468: “Thus, the CP concentration used in this study was not enough to significantly alter DNA integrity, but in combination with CXB becomes an effective anticancer drug.” This is speculative and such conclusion cannot be drawn from just gel based resection assays. The authors must try to show this in vivo using some kind of nuclease or DNA degradation assay.

Reviewer #2: The manuscript proposes use of Synergistic celecoxib and dimethyl-celecoxib combinations block cervix cancer growth without affecting the viability as there was no observed apoptosis or necrosis.

I feel that some cell biological analysis and control experiments could be added to improve the current manuscript. Overall, the paper is interesting.

1. The lack of apoptosis and necrosis was observed in the combo-treatment through FACS which is not sufficient to rule out cell death. Cleaved CASP3, cleaved CASP1, pMLKL should also be observed. Ferroptosis should also be analyzed.

2. Cell proliferation was analyzed by Ki67 however that would be not correct interpretation and EdU staining will strengthen the data.

3. The authors should mention the method of for viability assay in more details. They should include the information about the time course and if the cells detached were also collected for FACS analysis.

4. The author should provide explanation for why the significant decrease in ATP did not affect the viability as ATP would be critical for several biological processes.

5. The authors should provide statistical analysis in conventional terms by using *** rather than “abc” to avoid confusion for readers.

6. PLOS authors have the option to publish the peer review history of their article (what does this mean?). If published, this will include your full peer review and any attached files.

Reviewer #1: No

Reviewer #2: No

---

## [Author Response · Author response to Decision Letter 0]

13 Jun 2024

REVIEWER 1

“Robledo-Cadena et al has reported data suggesting synergistic association between celecoxib and dimethyl-celecoxib when used as a combinatorial therapy against cervical cancer by blocking its growth. Although, the study proposes exploratory avenues and scope of combinatorial therapy against cervical cancer, it is clear that further experimentation is required to fully validate the results. Multiple cell-lines as well alternative approaches should be utilized and the discussion should highlight the results pertinent to this study.” 

R: As suggested by the reviewer, new experimentation was performed using SiHa, another cervical cancer cell line. It was assessed in SiHa cells the effect of celecoxib and dimethyl-celecoxib combinations on cellular proliferation (new figure 1B), cell cycle (new figure S2), energy metabolism (new figures 2B and 3B), ROS production (new figure 5B), apoptosis (new figure 6B), caspases content (new figure 6C), glycoprotein activity (new figure 7B) and mitophagy (new figure S 5C). Accordingly, the text throughout the manuscript´s sections was modified. The general conclusion derived from these new results confirmed that CXB and DMC and their combinations increase selectivity against cervix cancer cells for its ability to target multiple pathways such as energy supply, ROS production, DNA stability, and drug defense (p. 19, 3rd paragraph).

“Line 55-57: This statement is broad. Combinatorial therapy and its advancement in the field of cancer research is already known. The authors need to focus on why this study or to be more specific the findings of this study is promising chemotherapy strategy against cervical cancer growth compared to what is already known in the literature.” 

R: The statement was rewritten (lines 57-61), and the specific findings of this study were better described. It should be also noted that drug combination therapy was recognized as an emerging approach in the first paragraph of Introduction in the original manuscript.

“The introduction needs work. What point are the authors trying to convey in the two paragraphs remains unclear. It is also confusing that Ralph, S.J.; et al. already reported CXB and DMC combinations with PA or CP blocked OxPhos flux (>80%) and consequently cellular invasiveness, so why did the authors choose to further analyze on several different functions of human cervix HeLa cells? What was the hint that additional biochemical pathways were involved? Could the authors explicitly clarify this point? Important: All critical and key findings in this paper should be repeated in one additional cell lines like SiHa/C-33A to support the findings.” 

R: The Introduction section was modified (p. 4, 1st paragraph) as well as the beginning of the Results section (p. 9, 4 paragraph) focusing on explaining the main differences between published work by Robledo-Cadena et al., 2020 and the present study. 

Critical experimentation of this paper was also performed in SiHa cells, as suggested.

“All bar graphs should show data points and the error bars should be + and – both.”

R: All bar graphs in each figure of the manuscript were modified, as suggested.

“Fig1. A and B What happens when you add CXB/CP/PA or DMC/CP/PA? What is the control here? Non-treated or DMSO? Fig1. A and B can be merged.”

R: Indeed, the drug combinations of CXB/CP/PA or DMC/CP/PA for 24 h promoted significant abolishment of HeLa or SiHa cell proliferation (see Figure A in "response to Reviewers" letter). However, their toxic effect on proliferation was significantly lower in HeLa cells than the effect observed with CXB/CP or CXB/PA and similar to DMC/CP or DMC/PA treatment (Fig 1A in the main text of R.1 manuscript). In SiHa cells, toxic effect by the three-drugs was higher than CXB or DMC combinations (Fig 1B in R.1). In both cells, three-drug treatment did not affect cell viability (viability > 85% in attached cells). This last observation in SiHa cells suggested CXB/CP/PA as potential anticancer alternative. Hence, these preliminary results were briefly described in p. 19, 2nd paragraph. 

In all experiments (original and new) performed in the present study, 70% ethanol/30% DMSO -treated cells were used as control. The DMSO and ethanol concentrations added to cells do not affect cellular proliferation, viability or energy pathway fluxes [Mol Pharm. 2018,15:2151-2164; Front Oncol. 2022, 12: 1018137]. A sentence was added in p. 4, last paragraph to clarify this point.

Fig.1A and Fig B were merged as suggested as new figure 1A.

“Fig1. C All uncropped western images should be reported in the supple info.”

R: Plos One requires that all uncropped and original western blot images presented in any study should be submitted in a separate file which is available to anyone who requests it. Therefore, all uncropped western images were now re-submitted in a separate pdf file named “Western blots Original files”. 

“Fig1. C the single addition of PA also promoted the lowering of Ki67 (6 times vs. control) in HeLa cells, which did not correlate with its null effect on cell growth, why?”

R: The level of Ki67 is certainly low in PA treated cells despite cell growth was not affected (original Fig 1C, now S1Fig). In this regard, it has been reported that Ki67 levels in HeLa cells do not always correlate with the cell proliferation state [Elife. 2016;5:e13722. doi: 10.7554/eLife.13722]. These authors found that rather than promoting cell proliferation, Ki67 is related with heterochromatin organization acquiring moonlight (no canonical) functions; thus, they suggest that Ki67 may be a biomarker under high level of heterochromatin compaction, which mediates long-range interactions between different regions of the genome that are packaged into heterochromatin. It has been also shown that very low PA concentrations (0.1 µM/48 h) induce chromatin fusion and instability in mouse embryonic fibroblasts (MEFs) without affect cellular growth [Biochem Biophys Res Commun. 2011 404:615. doi: 10.1016/j.bbrc.2010.12.018]. Thus, low Ki67 levels in PA-treated cells could be the result of chromatin disorganization. To address this issue, a sentence discussing the low level in Ki67 content vs. proliferation in PA-treated HeLa cells was included in p. 10, 2nd paragraph. 

“Important: The expression levels should also be compared using additional techniques like qPCR.”

R: We agree that it is nowadays fashionable and common practice to assess mRNA levels as indicator of cell functions and phenotypes. However, it is well documented that the mRNA levels of many metabolic enzymes and transcriptional factors have no correlation with the respective protein contents and activities, neither with pathway fluxes and cell functions [Electrophoresis 1997, 18: 533; Nat Rev Genet, 2012, 13:227; Trends Biochem Sci 2015, 40:1; FEBS J 2016, 283:54; Mol Syst Biol 2016, 12:883; Arch Biochem Biophys. 2023, 739:109559]. It should be noted that it is generally assumed that a transcript generates an active protein, and that an active protein similarly affects pathway functioning. These are incorrect assumptions because regulatory mechanisms are in place at these different levels of biological complexity. Thus, to achieve a thorough understanding of the modifications undergone by the energy metabolism pathways in cancer cells, it is more informative to experimentally analyze the cellular processes at the functional level. Particularly in this study a systematic analysis was performed by analyzing the main fluxes of OxPhos and glycolysis pathways (Figs 2 and 3). 

“Fig2. A Why is the level of HKI in CXB treated is significantly elevated compared to the control? Moreover, levels of CXB/CP treated looks very similar to controls? This is confusing and needs to be repeated and clarified. The HKII in CXB/CP and CXB/PA treated lanes seems completely diminished. What is this telling us? Is HKII a better marker compared to HKI? Where is the western analysis for DMC/CP and DMC/PA for Glycolysis and OxPhos?”

R: Some additional experimentation was undertaken in order to clarify the HKI content under the different experimental conditions. Results were similar to those previously found for glycolytic protein contents (original figure 2A, now S3 Fig). Indeed, HKI was significantly elevated in CXB treated cells, whereas in CXB/CP treated cells HKI did not change. This drug-induced puzzling HKI content response requires further research to understand the mechanisms involved and it is beyond the scope of the present study. Similarly, the low/negligible HKII content level also requires further experimentation to be able to propose it as a biomarker. Description of these observations was improved (p. 10, 4th paragraph).

As pointed out and discussed above, the present study focused on assessing functional properties of proteins and cells. In consequence, the effect of DMC and its combinations in HeLa cells was mainly assayed on pathway fluxes rather than on protein levels. Likewise, the CXB and DMC effects were determined on energy metabolism pathway fluxes in SiHa cells.

“Overall, the western qualities are not very good. The Nucleolin loading standard almost seems over saturated. Lanes should be marled properly “

R: Western blot images were improved, as suggested.

“Although PA or CP alone significantly affected OxPhos enzyme contents (Fig 2A), the maximal inhibitory effect on the mitochondrial membrane potential (ΔΨm, 45-50%) (data not shown), Why? “

R: PA or CP alone significantly affected more than 80% OxPhos enzyme contents which correlated with the low OxPhos flux (<90%), but not with the lowering of 45-50% in ΔΨm (New S3 Fig) in HeLa cells. This result has been observed in cancer HepG2 permeabilized cells, in which cellular respiration was modified whereas ΔΨm remained unchanged, even when cytochrome c oxidase was inhibited with cyanide [Mitochondrion 2004, 4:271–278; Biochem. J. 2006, 396:573]. It should be noted that respiratory and OxPhos rates cannot establish a linear relationship with ΔΨm, since the magnitude of the H+ gradient across the inner mitochondrial membrane obeys a logarithmic ratio of its out/in concentrations. In addition, it is recalled that OxPhos pathway requires a threshold ΔΨm value of around -80 mV to be able to initiate ATP synthesis. This issue was further explained in p. 11, 2nd paragraph.

“Fig3. A and B can be merged and should be presented with Fig2. B and C.”

R: Figures 3A and B were merged, but for clarity avoiding data overcrowding they were separated from other figures.

“Fig4. BNIP-3 (10-100 times) increase? This range does not make sense and needs to be re-visited.”

R: New experimentation with mitophagy protein contents was performed and results were re-analyzed and corrected. We thank the reviewer for pointing out this mishap. 

“Fig6. These experiments should be repeated with the usual chemiluminescent assay apoptosis or cell proliferation methods and the results should be correlated with what is now Fig6. A. Where is the result for DMC/CP and DMC/PA in Fig. 6B?”

R: We performed new experimentation for apoptosis and cell proliferation, analyzing (a) the levels of apoptotic mediated caspases (CASP1 and CASP3) (New Fig 6C) and (b) determining the cell cycle progression by flux cytometry (New S2 Fig) in HeLa and SiHa cells exposed to CXB or DMC and its combinations. The new data added further support to our previous results, indicating the lack of apoptosis in drug combination treatments (New Figs 6A and 6B). It was found that the CASP1 and CASP3 levels were negligible in drug-treated cells, indicating that caspase-induced cell death was not taking place. Experimentation with doxorubicin was included in order to demonstrate anti-CASP1 and -CASP3 antibodies effectivity. For proliferation analysis, the cell cycle progression in both HeLa and SiHa in the absence and presence of drug combinations was determined by FACS. This assay revealed that all drug combinations (CXB/CP, CXB/PA, DMC/CP, or DMC/PA) inhibited DNA replication and affect, apparently, cellular mitosis. These results were analyzed and included in p. 9, last paragraph, p. 10 1st paragraph, p. 12, last paragraph, p. 17, 3rd paragraph. 

New figures 6A, 6B, and 6C now display results that include DMC combinations.

“Important: It is not clear to this reviewer as to why the effect of CXB/CP and CXB/PA on P-glycoprotein content and activity was measured? What was the rationale? Is it clear at this point the MOA of these drugs that the authors considered testing for resistance? Where is the result for DMC/CP and DMC/PA in Fig. 7?” 

R: It is widely documented that P-glycoprotein is an ATP-dependent drug efflux pump for xenobiotic compounds [Chem Biol Interact. 2013, 203: 430; PLoS One. 2014, 9(6):e98804; Pharmacol Ther. 2015, 149:1]. This means that this protein couples the efflux of a wide variety of chemically and structurally different compounds to the hydrolysis of ATP and consequently, drug-efflux could be considered as a relevant ATP-dependent process. In this regard, low ATP levels produced by an inhibited OxPhos (c.f. Figs 3 and 4) could severely affect the protein functioning and could be considered as a potential CXB/DMC drug target. Description of these experiments’ rationale was improved (p.13, 1st paragraph). New experimentation with DMC was also described in p.13, 1st paragraph and in new figures 7A and 7B.

“Fig. 8 and 9 should be combined into one figure for better flow.”

R: Figures 8 and 9 (new Fig 8) were merged, as suggested.

“Not sure why the discussion and conclusion are separate? Please follow the submission guidelines.”

R: Discussion and conclusion were combined as submission guidelines indicate.

“Discussion: Line 467-468: “Thus, the CP concentration used in this study was not enough to significantly alter DNA integrity, but in combination with CXB becomes an effective anticancer drug.” This is speculative and such conclusion cannot be drawn from just gel based resection assays. The authors must try to show this in vivo using some kind of nuclease or DNA degradation assay.”

R: The method used for damaged DNA detection was omitted in the Material and Methods section from the original version, which was now included in the revised manuscript, in p. 8, last paragraph and p. 9, 1st paragraph. We thank the reviewer for pointing out this other mishap in our manuscript. 

In this study, we used the Gentra Puregene kit from QUIAGEN, which shows efficient extraction of DNA, successful removal of contaminants, production of sufficient amount of DNA for downstream workflow, and isolation of high quality and high purity DNA [Cancer Cytopathology, 2017, 125: 178-187. https://doi.org/10.1002/cncy.21812]. However, in agreement with the reviewer other DNA extraction methods should be performed in vivo to firmly support our previous conclusion, since DNA integrity could not be conclusively revealed by a single approach. Therefore, the mentioned statement was toned-down (p. 19, 1st paragraph). 

Reviewer #2: 

The manuscript proposes use of Synergistic celecoxib and dimethyl-celecoxib combinations block cervix cancer growth without affecting the viability as there was no observed apoptosis or necrosis.

I feel that some cell biological analysis and control experiments could be added to improve the current manuscript. Overall, the paper is interesting.

“1. The lack of apoptosis and necrosis was observed in the combo-treatment through FACS which is not sufficient to rule out cell death. Cleaved CASP3, cleaved CASP1, pMLKL should also be observed. Ferroptosis should also be analyzed.”

R: We performed new experimentation analyzing the levels of CASP1 and CASP3 in HeLa and SiHa cells exposed to CXB or DMC and its combinations (New Figure 6C). For CASP1 and CASP3, we found that the levels of both proteins were completely absent in drug-treated cells, indicating that caspase-induced cell death is not taking place. Experimentation with doxorubicin was included in order to demonstrate anti-CASP1 and -CASP3 antibodies effectivity. These results were analyzed and included in p. 12, 2nd and 3rd paragraphs, p. 17, 3rd paragraph. Ferroapoptosis activation by CXB or DMC or 

---

## [Decision Letter · Decision Letter 1]

16 Jul 2024

Synergistic celecoxib and dimethyl-celecoxib combinations block cervix cancer growth through multiple mechanisms

PONE-D-24-03423R1

Dear Dr.Rodríguez-Enríquez ,

We’re pleased to inform you that your manuscript has been judged scientifically suitable for publication and will be formally accepted for publication once it meets all outstanding technical requirements.

Kind regards,

Subhadip Mukhopadhyay, PhD

Academic Editor

PLOS ONE

Additional Editor Comments (optional):

Reviewers' comments:

Reviewer's Responses to Questions

**Comments to the Author**

1. If the authors have adequately addressed your comments raised in a previous round of review and you feel that this manuscript is now acceptable for publication, you may indicate that here to bypass the “Comments to the Author” section, enter your conflict of interest statement in the “Confidential to Editor” section, and submit your "Accept" recommendation.

Reviewer #1: All comments have been addressed

Reviewer #2: All comments have been addressed

2. Is the manuscript technically sound, and do the data support the conclusions?

Reviewer #1: (No Response)

Reviewer #2: Yes

3. Has the statistical analysis been performed appropriately and rigorously? 

Reviewer #1: (No Response)

Reviewer #2: Yes

4. Have the authors made all data underlying the findings in their manuscript fully available?

Reviewer #1: (No Response)

Reviewer #2: Yes

5. Is the manuscript presented in an intelligible fashion and written in standard English?

Reviewer #1: (No Response)

Reviewer #2: Yes

6. Review Comments to the Author

Reviewer #1: (No Response)

Reviewer #2: The authors have successfully addressed all of my questions by extensive new experiments. I have no other comments to make but to recommend the publication of the manuscript as corrected in this version.

7. PLOS authors have the option to publish the peer review history of their article (what does this mean?). If published, this will include your full peer review and any attached files.

Reviewer #1: No

Reviewer #2: No

---

## [Editor Report · Acceptance letter]

22 Jul 2024

PONE-D-24-03423R1 

PLOS ONE

Dear Dr. Rodríguez-Enríquez, 

I'm pleased to inform you that your manuscript has been deemed suitable for publication in PLOS ONE. Congratulations! Your manuscript is now being handed over to our production team.

Kind regards, 

on behalf of

Dr. Subhadip Mukhopadhyay 

Academic Editor

PLOS ONE